# A synaptic corollary discharge signal suppresses midbrain visual processing during saccade-like locomotion

Mir Ahsan Ali [1,6], Katharina Lischka [1,6], Stephanie J. Preuss [2,4], Chintan A. Trivedi[2,5] & Johann H. Bollmann [1,2,3] ✉

In motor control, the brain not only sends motor commands to the periphery, but also generates concurrent internal signals known as corollary discharge (CD) that influence sensory information processing around the time of movement. CD signals are important for identifying sensory input arising from self-motion and to compensate for it, but the underlying mechanisms remain unclear. Using whole-cell patch clamp recordings from neurons in the zebrafish optic tectum, we discovered an inhibitory synaptic signal, temporally locked to spontaneous and visually driven locomotion. This motor-related inhibition was appropriately timed to counteract visually driven excitatory input arising from the fish's own motion, and transiently suppressed tectal spiking activity. High-resolution calcium imaging revealed localized motor-related signals in the tectal neuropil and the upstream torus longitudinalis, suggesting that CD enters the tectum via this pathway. Together, our results show how visual processing is suppressed during self-motion by motor-related phasic inhibition. This may help explain perceptual saccadic suppression observed in many species.

What signals run through the brain when a small, brisk movement, such as a saccade of the eyes, makes our gaze jump from one point to another? First, thousands of photoreceptors across the retina generate signals driven by the abrupt, global shift of the retinal image; the retinal circuitry further processes these signals and sends them to higher sensory areas[1]. Simultaneously, a second type of signal can be observed in areas downstream of the retina, occurring around the time of the rapid eye movement. These movement-associated signals, known as corollary discharge (CD), encode variables such as the timing, strength, or direction of self-generated movements[2–6]. As a consequence, the nervous system can use CD signals as a flag to identify sensory signals arising from self-motion (reafference)[4] and account for them when processing and evaluating signals arising from external stimuli (exafference), which are ethologically more relevant.

Among various experimental demonstrations of CD signaling[2,6–10], recordings in the primate visual system show that the spiking activity of visual neurons both in visual cortical and subcortical areas[11–14] is transiently reduced during a saccade. This brief suppression likely contributes to the perceptual phenomenon of saccadic suppression, that is, the reduced perceptual sensitivity during saccadic eye movements. Another example is the fly visual system: here, during flight, visual neurons use CD signals in a subtractive computation to remove the visual input components caused by self-motion that would otherwise activate stabilizing visuomotor reflexes during flight[15,16], while, during walking, motor-related signals in these neurons may have course-stabilizing effects[17]. Collectively, experimental evidence has demonstrated that visual neurons are modulated during fast, saccade-like movements. In the vertebrate

[1]Developmental Biology, Institute of Biology I, Faculty of Biology, University of Freiburg, 79104 Freiburg, Germany. [2]Max Planck Institute for Medical Research, 69120 Heidelberg, Germany. [3]Bernstein Center Freiburg, University of Freiburg, 79104 Freiburg, Germany. [4]Present address: Springer Nature Group, Heidelberg, Germany. [5]Present address: Dept Cell and Developmental Biology, University College London, London, UK. [6]These authors contributed equally: Mir Ahsan Ali, Katharina Lischka. ✉e-mail: johann.bollmann@bio.uni-freiburg.de

visual system, however, the underlying cellular and synaptic mechanisms remain poorly understood.

The zebrafish model offers new opportunities to investigate in the intact brain where and how motor-related signals modulate visual processing during intermittent, 'saccade-like' locomotory sequences[18]. For locomotion, larval zebrafish typically use discrete swim bouts lasting 100-300 ms, interspersed with resting phases of 0.5 s to 1 s[19], much resembling the temporal characteristics of saccadic eye movement sequences when a primate scans a visual scene. Larval zebrafish are capable hunters: they track and capture moving prey in a seconds-long, goal-driven sequence of visually guided swim bouts[20–22], wherein the onset of the next swim bout depends on appropriate visual feedback immediately following the previous one[22,23]. As the image of the surrounding world sweeps across the retina during each swim bout, a barrage of afferent inputs is expected to reach retinorecipient areas, likely obscuring signals that encode local, ethologically relevant visual stimuli, or inappropriately feeding into visual reflex pathways such as the optomotor response. Therefore, a CD-based transient suppression of visual sensitivity while it is moving should be advantageous for the larva: it would help filter out signals from the self-motion generated blur on the retina, and potentially sensitize the visual system for detecting local visual stimuli immediately after the swim bout. While in recent years, much has been learned about the visual response properties of neurons in the intact zebrafish brain[24], information about motor-related signals in retinorecipient areas is lacking.

Here, we discovered a swim-related CD signal in the main retinorecipient center, the optic tectum (homologous to the mammalian superior colliculus) during spontaneous and visually evoked swim bouts in larval zebrafish. Using targeted patch clamp recordings[25–27] in combination with bilateral tail motor nerve recordings[28,29], we found that many tectal neurons receive a phasic inhibitory synaptic input, temporally locked to the swim bout. Its timing matches that of excitatory input in the same cells in response to abrupt large-field visual motion stimuli associated with self-motion. Furthermore, we show that this inhibitory signal is sufficient to suppress visually evoked spike output from tectal neurons during visually driven fictive swimming. Using rapid Ca²⁺ imaging, we show that short-lasting Ca²⁺ signals in specific layers of the tectal neuropil and in cell bodies of the torus longitudinalis (TL) located upstream of the tectum occur shortly after the onset of spontaneous swims, providing evidence that the TL-projection to the superficial neuropil is a likely entry point for inhibitory CD to the optic tectum. In summary, our results demonstrate an effective CD mechanism in the developing visual system capable of transiently suppressing visual information processing during saccade-like locomotion.

## Results

### Patch-clamp recordings reveal a motor-related voltage signal in tectal neurons

In the fully crossed retinotectal pathway of zebrafish, stimuli of different ethological relevance, such as small prey-like particles or dark expanding discs, generate different patterns of neuronal activity contralateral to the stimulated eye (Fig. 1a). These patterns are classified in intratectal circuitry, resulting in different patterns of output activity that are transmitted to downstream premotor areas. Here, they are translated into motor commands encoding different classes of swim bouts, directed toward or away from the visual object (Fig. 1a). Whether visual processing in the tectum is influenced by motor-related signals is, however, unknown. To investigate the impact of motor activity in the tectum, we performed two-photon-targeted patch-clamp recordings[30] from single tectal neurons and simultaneously recorded fictive motor activity bilaterally from the motor nerves of the tail muscles (Fig. 1b). We performed these recordings in the transgenic line *Tg(pou4f1-hsp70l:GFP)*[31]. In this line, GFP is expressed in neurons projecting to the ipsilateral reticular formation, the contralateral tectal

hemisphere, and local interneurons[32]. We recorded from both GFP-positive and GFP-negative cells in this line (see below). Furthermore, we visualized the dendritic branching pattern of the recorded neuron using sulforhodamine labeling (Fig. 1c).

To search for possible CD signals in the tectum, we measured the membrane potential of individual neurons in current clamp while the larva performed spontaneous fictive swim bouts. Notably, we often observed a small, brief hyperpolarization occurring after the onset of a swim bout (Fig. 1d). We calculated the swim-triggered averages of voltage recordings in 21 cells, in which we measured membrane voltage while the larva exhibited spontaneous fictive swimming behavior. When aligned to swim onset, we observed a transient hyperpolarization of −1.57 ± 0.32 mV ($n$ = 21 cells, mean ± SEM; Fig. 1e). The swim-related hyperpolarization was variable across cells: the average amplitude varied between 0.16 mV and −5.12 mV. This finding indicates that neurons in the tectum are modulated during spontaneous swimming, suggesting there is a CD signal relaying motor information to this central visual processing center.

### During spontaneous swim bouts, tectal neurons receive motor-related phasic inhibition

To investigate the cause of these motor-related voltage fluctuations, we performed measurements of synaptic currents in tectal neurons in voltage clamp to resolve putative inhibitory postsynaptic currents (IPSCs). When held at holding potentials of 0–10 mV, we observed that in a majority of cells, transient outward currents occurred immediately after a spontaneous swim bout (Fig. 2a). The dependence of these currents on membrane voltage, with a negative reversal potential, and their rise and decay kinetics were consistent with ionotropic inhibitory synaptic conductances, likely mediated by GABAergic synapses (Supplementary Fig. 1a–d). The IPSCs were composed of a few individual current peaks and ended shortly after the swim bout. We calculated the charge integral as a measure of the strength of the synaptic input and measured the delay of the IPSC relative to swim onset (Fig. 2b). The charge histogram was multi-modal: it showed a peak around 0 pC (blue bars and dashed Gaussian fit in Fig. 2c), corresponding to cells in which no phasic IPSC was observed ($n$ = 24). 32 cells, however, received phasic inhibitory synaptic input >0.8 pC after a swim bout (blue bars overlaid with magenta color, Fig. 2c). The delay between the onset of swimming and the onset of the IPSC was 124 ms ± 5 ms (mean ± SEM, $n$ = 32; Fig. 2d).

The cells in which we had measured a motor-related hyperpolarization signal (Fig. 1e) are a subset of these 56 cells. Therefore, we could compare the average hyperpolarization measured in current clamp in each cell with the average amount of inhibitory charge transfer measured in voltage clamp (Fig. 2e). We found that the transient drop in membrane voltage was correlated with the IPSC charge. We conclude that tectal cells receive a fast CD signal in form of a strong, phasic inhibitory synaptic input, which causes transient membrane hyperpolarization and could therefore modulate tectal cell activity during self-generated locomotion.

### Fast inhibitory currents in tectal cells also occur during visually driven, directed swimming

Next, we investigated the extent to which motor-related CD signals also occur during visually evoked fictive swimming behavior. To do so, we combined single-cell patch clamp recordings with measurements of bilateral motor activity while presenting the larva with different visual stimulus patterns (Fig. 3a). Previous work showed that both freely swimming and tethered larvae perform target-directed approach swims in response to small moving stimuli whereas large moving objects or expanding discs preferentially evoke escape swims[21,22,33,34]. Here we observed that also in the fictive swim preparation, larvae generated corresponding swim patterns when presented with these stimulus types. Notably, visually evoked swims were also

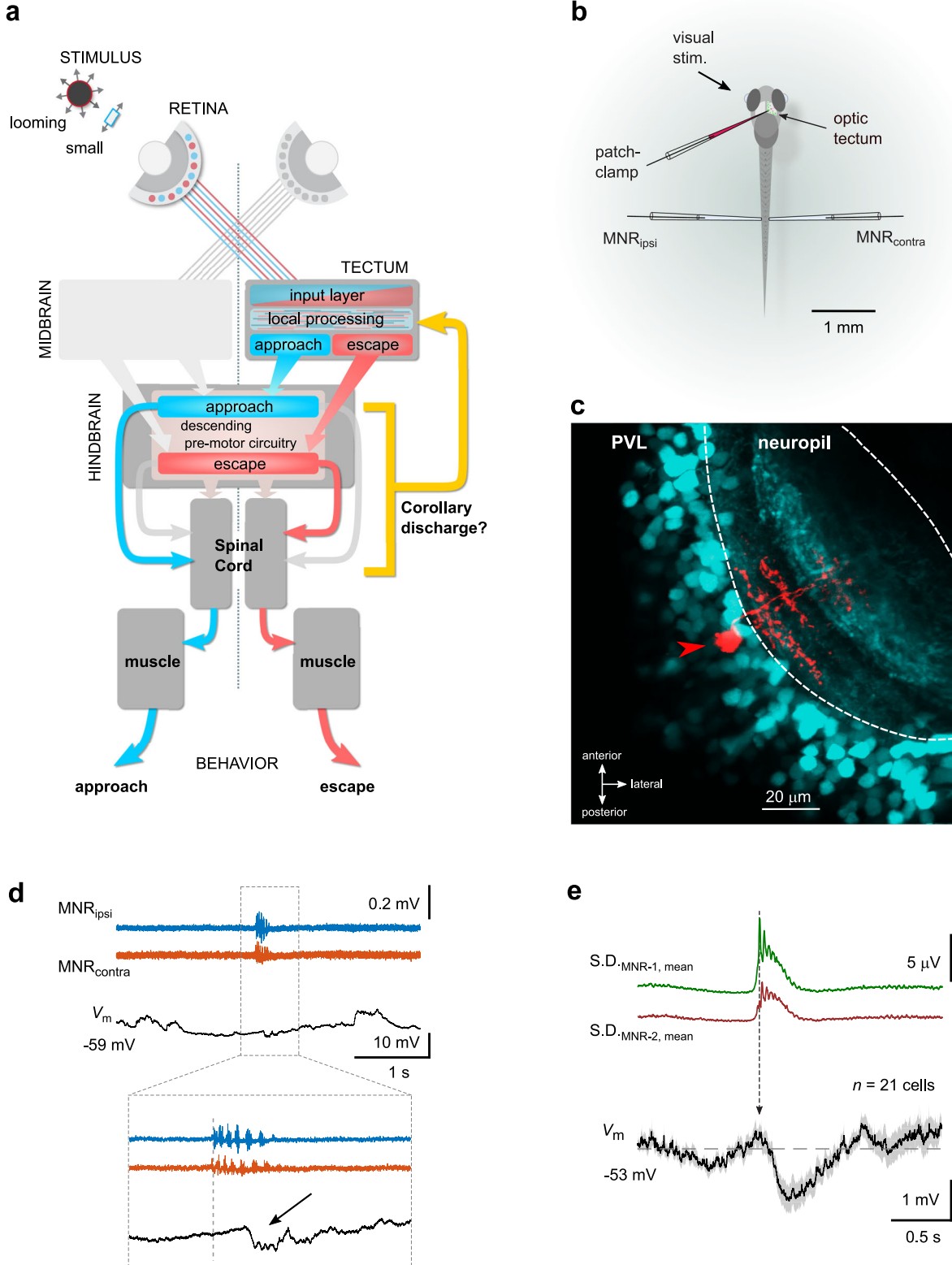

accompanied by brief inhibitory synaptic currents, similar to those occurring during spontaneous swims (Fig. 3b). To elucidate whether larvae in the fictive swim preparation exhibited regular goal-directed swimming behavior, we used the bilaterally measured motor nerve activity to calculate the summed swim power and a direction index for each swim, as a proxy for the intended swim direction (Fig. 3c). This analysis showed that swims evoked by escape-inducing stimuli (large

objects, looming discs), elicited swims with larger swim power than those occurring spontaneously or evoked by small moving objects (Fig. 3d). Importantly, swims evoked by small prey-like stimuli had a positive direction index, indicating appetitive, target-directed swimming, whereas those swims evoked by large or looming stimuli had negative direction indices (Fig. 3e). This strongly supports the notion that larvae in the fictive swim preparation exhibit normal swimming

**Fig. 1 | Patch-clamp recordings reveal a motor-related voltage signal in tectal neurons. a** Schematic of the visuomotor pathway in larval zebrafish, including the retina, the optic tectum, hindbrain premotor circuits, spinal cord and axial tail muscles. Visual stimuli of different behavioral value (e.g. loom vs small, prey-like) are processed in the contralateral tectal hemisphere and trigger aversive (escape, red) or target-directed swims (approach, blue) through parallel pathways into the hindbrain premotor circuitry. Putative swim-related feedback signals may affect tectal visual processing during phases of swimming (corollary discharge, yellow). **b** Recording configuration. Whole-cell patch clamp recording from a tectal neuron is combined with bilateral extracellular field recordings from tail motor nerves ($MNR_{ipsi}$ and $MNR_{contra}$). **c** Recorded tectal neuron (red) labeled with sulforhodamine-B via the patch pipette in *Tg(pou4f1-hsp70l:GFP)* transgenic

background (cyan). PVL periventricular cell body layer. Micrograph representative of 56 independent experiments. **d** Example recording of membrane voltage ($V_m$, black) in a tectal neuron during a spontaneous fictive swim bout ($MNR_{ipsi}$, $MNR_{contra}$, blue and red, respectively). Inset: magnified view of swim event (onset indicated by dotted line). Note the associated short hyperpolarization in the voltage trace (arrow). **e** Swim-triggered average of voltage signals in tectal neurons (bottom solid trace: mean; shaded area: ± SEM; average across 21 individual, baseline-subtracted cell averages). $V_m$ traces were aligned by the onset of swimming (vertical arrow), measured as the first burst in a fictive swim bout. Top: Standard deviation traces of motor nerve recordings (average from 330 swim events). Source data are provided as a Source Data file.

behavior, which much resembles that of freely swimming or tethered larvae.

Next, we determined the synaptic charge transfer and delays of IPSCs for the different stimulus categories. For loom-evoked escape swims, we observed a synaptic charge transfer significantly larger than that in the other stimulus categories (Fig. 3f), reflecting the higher swim power generated by the larva during this swim type (Fig. 3d). The IPSC delays did not differ significantly for different stimulus classes (Fig. 3g). Overall, the timing of the motor-related CD signal was relatively invariant. To test whether the strength of the inhibitory input correlated in any way with the motor activity measured on either side of the tail, we performed a multi-regression analysis of IPSC charge with measured unilateral swim power (Fig. 3h). This analysis showed that charge transfer (measured in tectal cells contralateral to the stimulated eye) was positively correlated with swim power on the same side of the tail, contralateral to the stimulus (Fig. 3h, see also Supplementary Fig. 2 for an extended analysis). In summary, we conclude that generally, any swim activity is accompanied by short inhibitory CD-like signals in the tectum and that the strength of this CD signal is positively correlated with the recruitment of muscular activity on the same side.

**Swim-related inhibition suppresses visually evoked spike output**
If the CD signals observed here serve to suppress the processing of reafferent input that results from self-generated movement, then we expect to see an effect on visually evoked spiking activity. To test this, we used expanding discs as a stimulus because they elicited a transient increase in firing rate in many tectal neurons (Fig. 4a, b). We observed both an increase of the spike count, summed across the population of recorded cells (Fig. 4b), and depolarization of the membrane potential (after spikes were digitally removed; Fig. 4c) during looming stimuli. In addition, the looming disk stimuli evoked fictive escape swims (Fig. 4d).

Notably, in many cells, the firing rate was briefly reduced during a fictive swim burst or began to rise only afterward (Fig. 4a, i, ii). In other cells, swimming activity did not appear to affect firing rate (Fig. 4a, iii). To further address this, we determined the influence of swimming activity on firing rate by aligning looming-evoked firing rate changes relative to the swim onset (Fig. 4e). The time course of the firing rate relative to swim onset was variable for different cells (Fig. 4e): some cells started to fire before or during the swim bout, others started only afterwards. However, when aligned to the swim, the spike count summed over all cells showed a clear drop immediately after swim onset (Fig. 4f). The effect of motor activity on firing rate became clearer when we divided the data into two groups: those cells that received an appreciable inhibitory synaptic input during spontaneous swimming activity (a subset of 10 cells from those in Fig. 2c (magenta bars), with charge input >0.8 pC), and those that did not receive appreciable inhibitory input (a subset of 11 cells from those in Fig. 2c (blue bars) with IPSC charge <0.8 pC). The spike count summed across the group of synaptically inhibited cells was strongly reduced during a swim bout (Fig. 4g, magenta bars), whereas that for the group without synaptic inhibition remained nearly constant (Fig. 4g, blue bars, see

also Supplementary Fig. 3a). Moreover, the time course of the membrane potential (after removal of spikes) exhibited a significant transient drop during swimming activity only in cells receiving synaptic inhibition, but not in those without inhibitory synaptic currents (Fig. 4h, i; see also Supplementary Fig. 3b, c for details). We did not observe a dependence of whether or not a cell received motor-related inhibition with the position of the neuron, as cells from the two groups were found at about the same central region of the tectal cell body layer which we recorded from (Supplementary Fig. 3d, e). In summary, we found that an inhibitory CD signal briefly suppresses visually driven activity in many tectal cells, which suggests that a neuronal correlate of motor-related suppression, or 'saccadic suppression' in a broader sense, is implemented in the tectal circuitry.

**Swim-related inhibitory CD signals are timed to cancel excitatory inputs evoked by fast whole-field motion stimuli**
If the CD signals observed here serve to specifically suppress those excitatory signals in the tectum that arise from the abrupt shift of the retinal image during a swim bout (reafference), then this excitation and the inhibitory CD signal must coincide in time. To test this, we measured the time course of excitatory postsynaptic currents (EPSCs) evoked by a wide-field motion stimulus, as would be expected during an abrupt, discrete movement (Fig. 5a). During forward swimming, optic flow sweeps across the eye in the front-to-back direction. We observed that a grating moving abruptly from front-to-back evoked phasic EPSCs in tectal neurons (Fig. 5b). The delay between movement onset of the stimulus and the excitatory current was 161 ms ± 8 ms (mean ± SEM, $n = 24$ cells, Fig. 5c). This is somewhat longer than the delays of ~75–150 ms measured for inhibitory CD signals (Fig. 2d). In a subset of these cells, we measured the delays of moving grating-evoked excitatory inputs and swim-related inhibitory inputs for a within-cell comparison (Fig. 5d–f). Inhibitory synaptic input preceded excitatory synaptic input in these cells by 70 ± 25 ms (mean ± SEM, Fig. 5f). We conclude that the motor-related CD signal is appropriately timed to shunt self-motion-generated reafferent excitation in the tectum.

**Spatial distribution of swim-related Ca$^{2+}$ signals in the tectal neuropil**
Next, we searched for evidence where in the tectal neuropil the phasic inhibitory CD signal may be transmitted. As feature- and task-related synaptic inputs are often localized in specific tectal layers[26,35,36], we examined whether there is evidence for a laminar organization of swim-related CD signals. We argued that presynaptic compartments that phasically release an inhibitory transmitter shortly after swim onset should exhibit a transient increase in [Ca$^{2+}$], whose rising phase represents the time of presynaptic action potential firing[37] and should hence start only after the swim onset (Fig. 6a). We therefore used two-photon Ca$^{2+}$ imaging at high spatial and temporal resolution in a transgenic line that expresses GCaMP5G pan-neuronally[38] and characterized in a central strip of the neuropil the distribution of Ca$^{2+}$ transients that occurred around the onset of

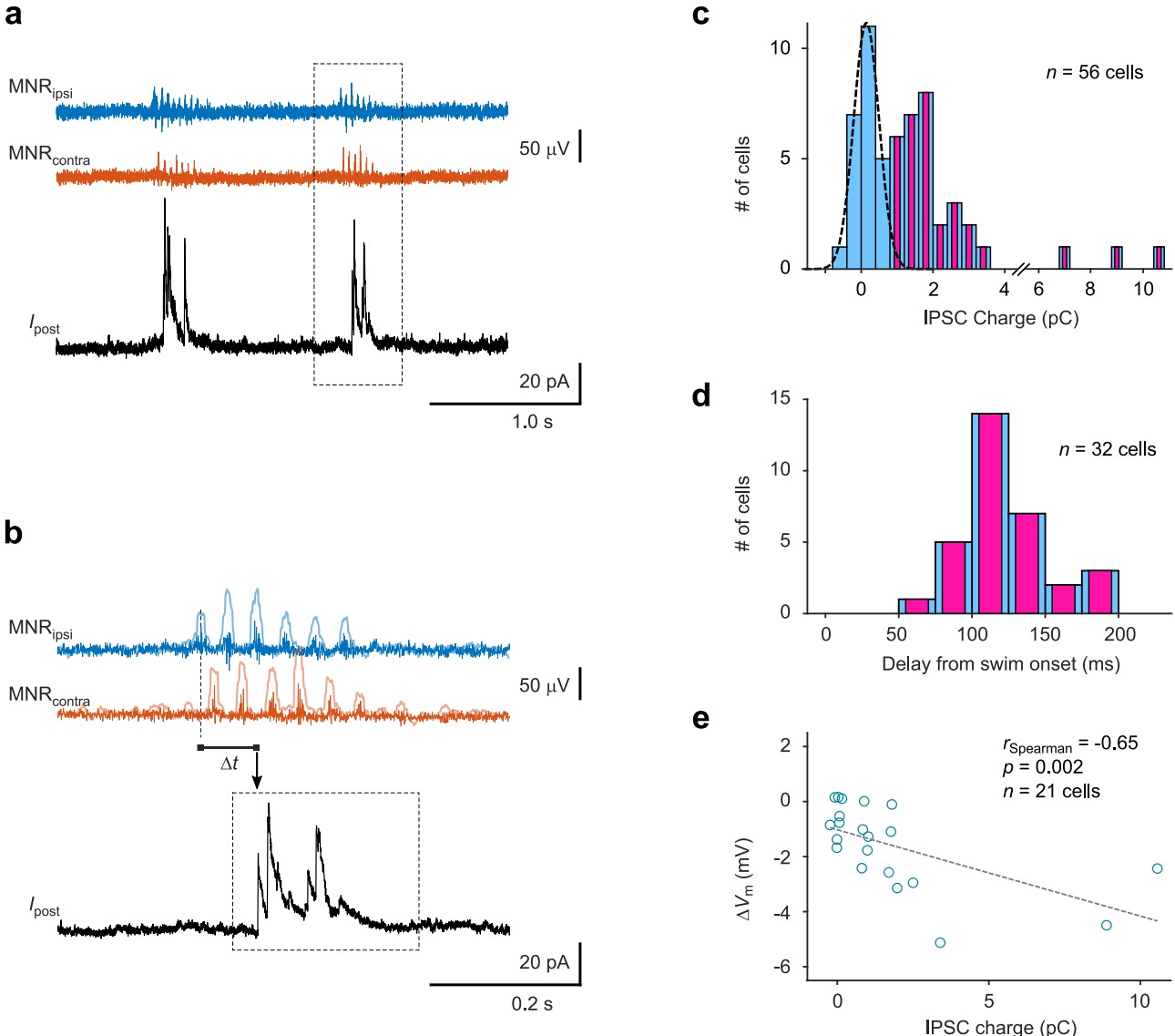

**Fig. 2 | During spontaneous swim bouts, tectal neurons receive motor-related phasic inhibition. a** Example recording of inhibitory membrane currents ($I_{post}$) during spontaneous fictive swim activity. Swim events (bursts in the $MNR_{ipsi}$ and $MNR_{contra}$ traces) are closely followed by short-lasting, large inhibitory post-synaptic currents (IPSCs) in a tectal neuron. Holding potential was +10 mV. **b** Magnified view of the second event from (**a**). IPSC charge was measured in a 250 ms window (dotted rectangle). Delay ($\Delta t$) was measured between the first burst in the swim bout and the IPSC onset. MNR traces are overlaid with traces representing their standard deviation (see Methods). **c** Histogram of inhibitory charge transfer in tectal neurons associated with spontaneous swimming (individual cell averages from $n = 56$ neurons). Dashed curve: Gaussian fit to the left peak of the charge histogram. In the majority of cells ($n = 32$), inhibitory charge transfer was

larger 0.8 pC (bars co-labeled in magenta), indicating non-negligible inhibitory swim-related input. **d** Histogram of delays between swim onset and IPSC onset. Individual cell averages from $n = 32$ cells with non-negligible inhibitory swim-related input. **e** Scatter plot of swim-related transient hyperpolarization measured in current clamp ($\Delta V_m$) and IPSC charge transfer measured in voltage clamp from a subset of cells in (**c**) where both modes of recording were applied ($n = 21$; 8 cells with IPSC charge <0.8 pC, 13 cells with charge >0.8 pC). $\Delta V_m$ and IPSC charge are negatively correlated ($r_{Spearman} = -0.65$, $p = 0.002$, Spearman rank correlation), which also holds if the two rightmost data points are excluded from correlation analysis ($p = 0.014$, see Supplementary Fig. 1e). Source data are provided as a Source Data file.

spontaneous swim bouts (Fig. 6a–c, see also Supplementary Fig. 4). We systematically subdivided the imaged area into small ROIs to identify localized $Ca^{2+}$ transients (Fig. 6c). The pattern of local $Ca^{2+}$ signals during single swims was remarkably heterogeneous (Fig. 6c and Supplementary Fig. 4a). Using a method for automatic detection of $Ca^{2+}$ transients in a window ±4 s around swim onset (Supplementary Fig. 4b), we found that $Ca^{2+}$ transients occurred throughout the neuropil, with variable onset times relative to the onset of swimming (Fig. 6c). The occurrence of $Ca^{2+}$ transients rose supralinearly before swim onset and dropped rapidly afterwards (Fig. 6d), consistent with the idea that the spontaneous build-up of tectal activity contributes

to the triggering of swim bouts[39]. Notably, however, 18% of all $Ca^{2+}$ transients began in a time window 50 to 350 ms after swim onset (Fig. 6d, yellow bars). We hypothesize that these post-swim $Ca^{2+}$ signals reflect at least in part activity in those presynaptic compartments that transmit the inhibitory CD signal. Therefore, we examined how active ROIs (corresponding to those, where a $Ca^{2+}$ transient was detected) were distributed in time and space across the neuropil (Fig. 6e). The distribution was remarkably non-uniform: active ROIs were clustered around the time of swim onset in the deep neuropil (depth level 0-20%, containing the *stratum album centrale*, SAC), and in the most superficial layer (depth level 90-100%, containing the

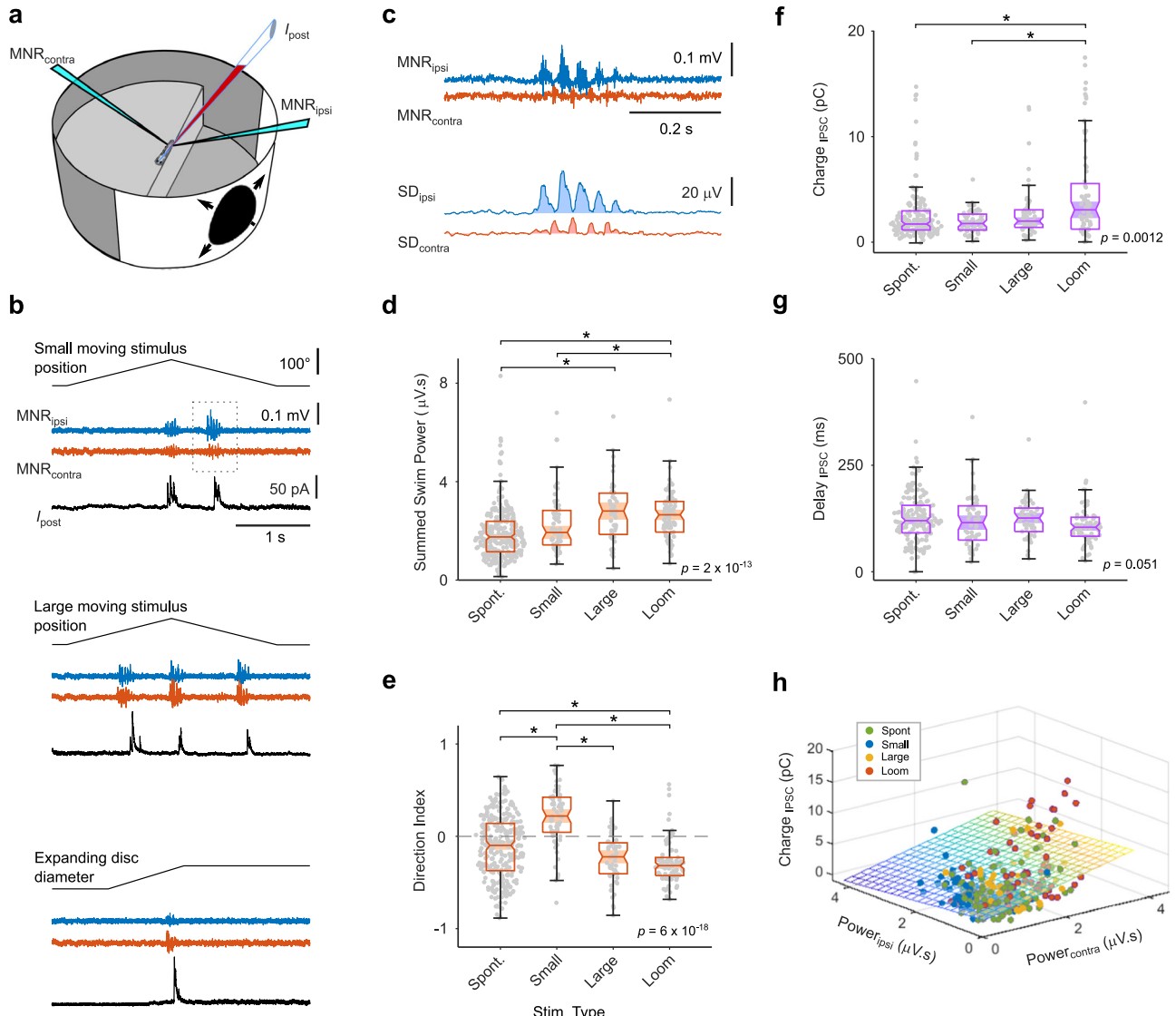

**Fig. 3 | Fast inhibitory currents in tectal cells during visually driven, directed swimming. a** Recording configuration. Visual stimuli are projected on the side wall of the cylindrical arena. **b** Simultaneous bilateral recording of motor nerve activity ($MNR_{ipsi,contra}$) and patch clamp recording from a tectal cell ($I_{post}$) during different, visually evoked swims. All traces from same neuron. **c** Motor nerve recording during small stimulus presentation (from rectangle in **b**) exhibits stronger activity on the side ipsilateral to the stimulus. Lower traces: standard deviation of motor nerve recording (10 ms moving window). Shaded areas indicate swim power on the ipsi- (blue) and contralateral (red) side. **d** Sum of ipsi- and contralateral swim power is different for spontaneous and visually evoked swims ($p = 2 \times 10^{-13}$, Kruskal-Wallis test, $n = 493$ bouts from 56 larvae). Swims in response to large rectangles and looms are stronger than spontaneous swims (Large vs. Spont.: $p = 2 \times 10^{-7}$; Loom vs. Spont.: $p = 2 \times 10^{-10}$; Loom vs. Small: $p = 0.017$). **e** Directional indices of spontaneous and visually evoked swims exhibit significant differences consistent with stimulus type ($p = 6 \times 10^{-18}$, Kruskal–Wallis test). Small vs. Spont.: $p = 5 \times 10^{-10}$; Loom vs. Spont.: $p = 8 \times 10^{-5}$; Large vs. Small: $p = 8 \times 10^{-11}$; Loom vs. Small: $p = 8 \times 10^{-18}$. Same data as in **d**. **f** Inhibitory charge transfer during spontaneous and visually evoked swims

differs between swim types ($p = 0.0012$, Kruskal–Wallis test). Data from recordings of cells with non-negligible charge transfer (>0.8 pC, magenta cells in Fig. 2c; 345 events from 32 cells). Loom vs. Spont.: $p = 0.003$; Loom vs. Small: $p = 0.0027$. **g** Delays between swim onset and IPSC onset for spontaneous and visually evoked swims exhibit no significant differences ($p = 0.051$). Statistical differences between groups in panels **d**–**g** were evaluated using Kruskal–Wallis tests with post-hoc pairwise comparison using Tukey-Kramer method for multiple comparisons. Box-and-whisker plots in panels **d**–**g** indicate the median, and upper and lower quartiles (box edges). Whiskers: upper and lower limit of data range, up to a maximum of 1.5x interquartile range. **h** Scatter plot of IPSC charge associated with different swim types. Colored plane represents multiple regression model of charge transfer as a function of swim power in the ipsi- and contralateral motor nerve recording (F-test for multiple regression model: $R^2 = 0.11$; F-statistic = 20.6; $p = 3.5 \times 10^{-9}$). Regression coefficient for contralateral swim power is significantly different from 0 (t-statistic = 6.42, $p = 4.4 \times 10^{-10}$), but not for ipsilateral swim power (t-statistic = −1.81, $p = 0.07$). See also Supplementary Fig. 2. Source data are provided as a Source Data file.

*stratum marginale* (SM) and *stratum opticum* (SO)). In the SAC, in which the proximal axon segments of tectal projection neurons are packed, the number of active ROIs peaked in the interval [−0.15 s; −0.05 s], immediately before swim onset, consistent with the notion that tectal premotor activity is transmitted via this pathway and contributes to triggering a spontaneous swim. By contrast, in the SM/SO the number of active ROIs peaked in the interval [−0.05 s; 0.05 s]

and exhibited relatively high activity in the interval [0.05 s; 0.25 s] shortly thereafter. Given the asymmetrical shape of the temporal distribution (Fig. 6d), we consider active ROIs in the interval [−1.05; +0.35 s] as likely related to the occurrence of the swim bout. Therefore, we calculated the cumulative distribution of $Ca^{2+}$ transients in this interval, pooled according to the stratification of the tectal neuropil. This showed that the onset of $Ca^{2+}$ signals in the SM/SO and

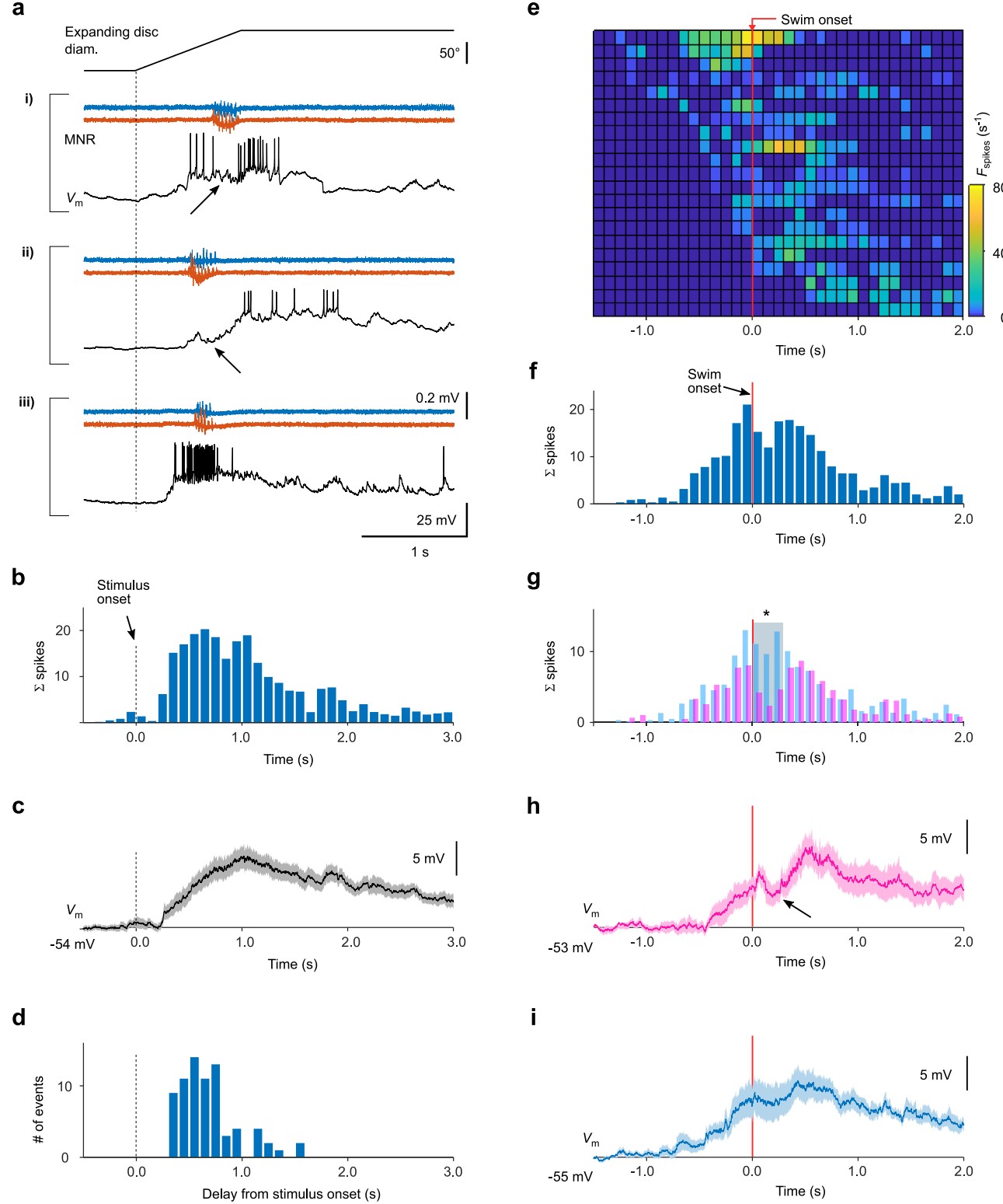

the deep SAC rose and fell mainly around swim onset, whereas those in the more central layers containing the *stratum fibrosum et griseum superficiale* (SFGS) and *stratum griseum centrale* (SGC) were more uniformly distributed (Fig. 6f). We compared the relative fraction of 'post-swim' $Ca^{2+}$ transients starting in the interval [0.05 s; 0.35 s] after swim onset relative to 'pre-swim' ROIs in the interval [−1.05 s; −0.05 s] for each of these neuropil regions to that across all layers (Fig. 6g). Notably, in the SM/SO layer, $Ca^{2+}$ transients occurred significantly more often in the post-swim interval compared to the occurrence

across all layers (temporal shuffling of the fluorescence data relative to swim onset removed this bias; see Supplementary Fig. 4c, d). Most types of tectal neurons do not extend neurites to this very superficial neuropil region. Instead, it is strongly innervated by afferent fibers from the mediodorsal torus longitudinalis (TL)[40,41], making the latter a candidate pathway for relaying CD signals from premotor areas to the tectal circuitry. Therefore, in the next step, we set out to investigate the temporal profile of neural activity in this region upstream of the tectum in relation to spontaneous swim events.

**Fig. 4 | Swim-related inhibition transiently suppresses visually evoked spike output. a** Motor nerve recordings (MNR) of looming-evoked swim events and simultaneously recorded spiking activity ($V_m$) in three different neurons (i–iii). Stimulus onset indicated by vertical dashed line. **b** Population spike time histogram, evoked by looming stimuli. Spikes were counted in 100 ms bins, averaged across sweeps for each cell and then summed over all cells ($n = 21$ cells). **c** Membrane voltage change of the recorded cells in response to looming stimuli. Average across individual, baseline-subtracted cell averages ($n = 21$ cells, mean ± SEM). Spikes removed by interpolation between spike onset and offset. **d** Delay histogram of fictive swim events relative to stimulus onset. **e** Histogram of instantaneous spike rate, evoked by looming stimuli, aligned to swim onset. Each row represents spiking activity from one cell ($n = 21$). Rows in histogram vertically sorted according to onset of cell spiking relative to swim onset. **f** Population spike time histogram, aligned to swim onset (vertical red line), summed over all cells shown in **e**. **g** Population spike time histograms as in **f**, but plotted separately for cells with inhibitory charge transfer >0.8 pC (magenta), and those with negligible inhibitory charge (<0.8 pC, blue). The total spike count summed in a 300-ms window following swim onset (gray bar) is significantly smaller in the inhibited cell population (magenta) than in the population without inhibition (one-tailed comparison of measured vs predicted spike counts using Poisson statistics; $p = 4.6 \times 10^{-5}$). **h, i** Membrane voltage traces (spikes removed by interpolation), aligned to swim onset. Note the transient decrease in membrane voltage during swimming for cells receiving swim-related inhibition (arrow in panel **h**; average drop in $V_m = -2.8 \pm 0.9$ mV, $n = 10$ cells, $p = 0.014$, two-sided Wilcoxon signed-rank test), but not in cells with negligible charge transfer (panel **i**; average drop in $V_m = -0.6 \pm 0.9$ mV, $n = 11$ cells, $p = 0.7$). Traces show averages across individual, baseline-subtracted cell averages (mean ± SEM, $n = 10$ cells in **h**, $n = 11$ cells in **i**). See also Supplementary Fig. 3. Source data are provided as a Source Data file.

## Temporal distribution of swim-related $Ca^{2+}$ signals in the torus longitudinalis

For imaging neural activity in TL cells, we used the line *Tg(elavl3.1:Gal4-VP16;UAS:GCaMP3)*[26], in which cell bodies in the TL region, located anteriodorsally between the tectal hemispheres, expressed GCaMP3 (Fig. 7a, dashed outline and right inset). Combining $Ca^{2+}$ imaging with recordings of motor nerve activity, we observed that spontaneous swim bouts were often accompanied with $Ca^{2+}$ transients in individual TL neurons (Fig. 7b). Notably, the onsets of $Ca^{2+}$ transients in TL neurons started typically after swim onset (Fig. 7c) and were distributed within a few hundred ms after the swim bout (Fig. 7d). Out of 352 neurons that could be identified because of GCaMP-expression (in 17 scanned regions from 5 fish), 214 neurons (61%) exhibited post-swim $Ca^{2+}$ transients, consistent with the notion that a considerable subset of TL neurons mediate a transient post-swim signal to the tectal neuropil. Together with the finding that a large fraction of $Ca^{2+}$ transients in the tectal SM/SO occur in a short post-swim time window, these findings suggest that the most superficial neuropil is a key input layer for CD signals relayed from extratectal sources via the TL.

## Cells with diverse dendritic morphologies receive CD synaptic inhibition

We further searched for cues where in the tectal neuropil inhibitory CD signals may be transmitted. If CD signals enter the tectum in the SM/SO layer, do only those cells that extend dendrites all the way up into the most superficial neuropil receive motor-related inhibitory synaptic input? We examined this by comparing the dendritic profiles of recorded neurons with and without CD synaptic inhibition (Fig. 8a–d). GFP-positive cells recorded in the *Tg(pou4f1-hsp70l:GFP)* line had many neurites extending in the deep neuropil (Fig. 8a, b). They did not, however, arborize in layers more superficial than the central SFGS, making it unlikely that they received direct synaptic input from a CD-mediating afferent input in the SM/SO. Also, whether or not a cell received motor-related inhibition was unrelated to its dendritic profile (compare Fig. 8b, upper vs. lower panel). Interestingly, GFP-positive cells, which to a large part represent projection neurons[32], were not the only cells receiving CD inhibitory input. When patching GFP-negative neurons, we observed cells with markedly distinct morphology also receiving phasic motor-related inhibition (Fig. 8c, d). For example, cells appearing to be local interneurons, such as bistratified neurons with a dorsal dendritic branch near the SFGS/SO boundary, were among those with CD inhibitory input. Together, the finding that cells without dendrites anywhere near the superficial SM/SO still receive strong motor-related inhibition suggests that there is a type of inhibitory relay neuron in the tectum that converts an excitatory CD input into a local inhibitory signal and distributes it across one or more layers.

To further corroborate this, we investigated the distribution of inhibitory synapses in the neuropil. As fast inhibitory transmission in the teleost tectum is predominantly mediated by $GABA_A$ receptor channels, we used antibody labeling against the GABAR subunits β2/β3, which are strongly expressed in the tectal neuropil and localize in specific layers in several adult teleost species[42–44] (Supplementary Fig. 5). We argued that laminae in the neuropil that lack $GABA_A$ receptors could be excluded as layers in which strong inhibitory CD signals are transmitted. Contrary to that notion, we observed immunoreactivity against GABAR subunits β2/β3 across all layers, without a distinct laminar organization. Furthermore, GABAR β2/β3-subunits were also detected within the periventricular cell body layer (Supplementary Fig. 5d). The diffuse distribution of GABA receptors suggests that the transmission of inhibitory CD signals need not be restricted to one or a few specific layers but can in principle reach a variety of morphologically distinct tectal neurons in different layers of the neuropil.

Together, these observations suggest that motor-related inhibitory signals are distributed widely across the tectum, exerting a transient suppressive effect on multiple cell types with distinct functional roles.

## Discussion

We discovered a robust synaptic CD signal in visually driven neurons of the optic tectum. We observed this phasic inhibitory input, which was temporally locked to fictive motor activity, in many cells with different morphologies, both during spontaneous and visually evoked swim patterns. Strikingly, this inhibitory input suppressed action potential firing during discrete fictive swim bouts, supporting the notion that the processing of sensory activity arising from self-motion (reafference) is effectively suppressed at the level of the optic tectum. $Ca^{2+}$ imaging in the tectal neuropil revealed a large fraction of post-swim $Ca^{2+}$ signals in the most superficial layer, which together with the observation that many TL neurons exhibit bursts of activity shortly after spontaneous swims suggests that CD signals from premotor areas enter the tectum in this layer via TL projection neurons (Fig. 9). The results are significant as they suggest a promising model for investigating saccadic suppression, tractable at the cellular and synaptic level, in the retino-tectal pathway.

What could be the functional role of these inhibitory CD signals in tectal circuitry? Generally, CD signals inform sensory brain areas about ongoing self-generated movements and influence the processing of incoming sensory signals. In what way does the CD signal observed here influence ongoing visuomotor processing?

Apparently, the CD inhibitory input is effective in suppressing tectal output activity in the moment the larva swims in a saccade-like manner. This is because first we observed that in the *Tg(pou4f1-hsp70l:GFP)* line, GFP-positive neurons, whose axons contribute to a distinct output channel projecting from the tectum to hindbrain premotor circuits[32], receive a pronounced motor-related inhibitory input (Fig. 2) and exhibit transient suppression of spiking (Fig. 4). Second, the inhibitory CD signal was adequately timed to coincide with, and therefore counteract, the reafferent excitation evoked by global stimulation of the retina, expected to arise from the fish's own locomotion (Fig. 5). Third, since the CD signal occurred reliably during various stimulus conditions and as its

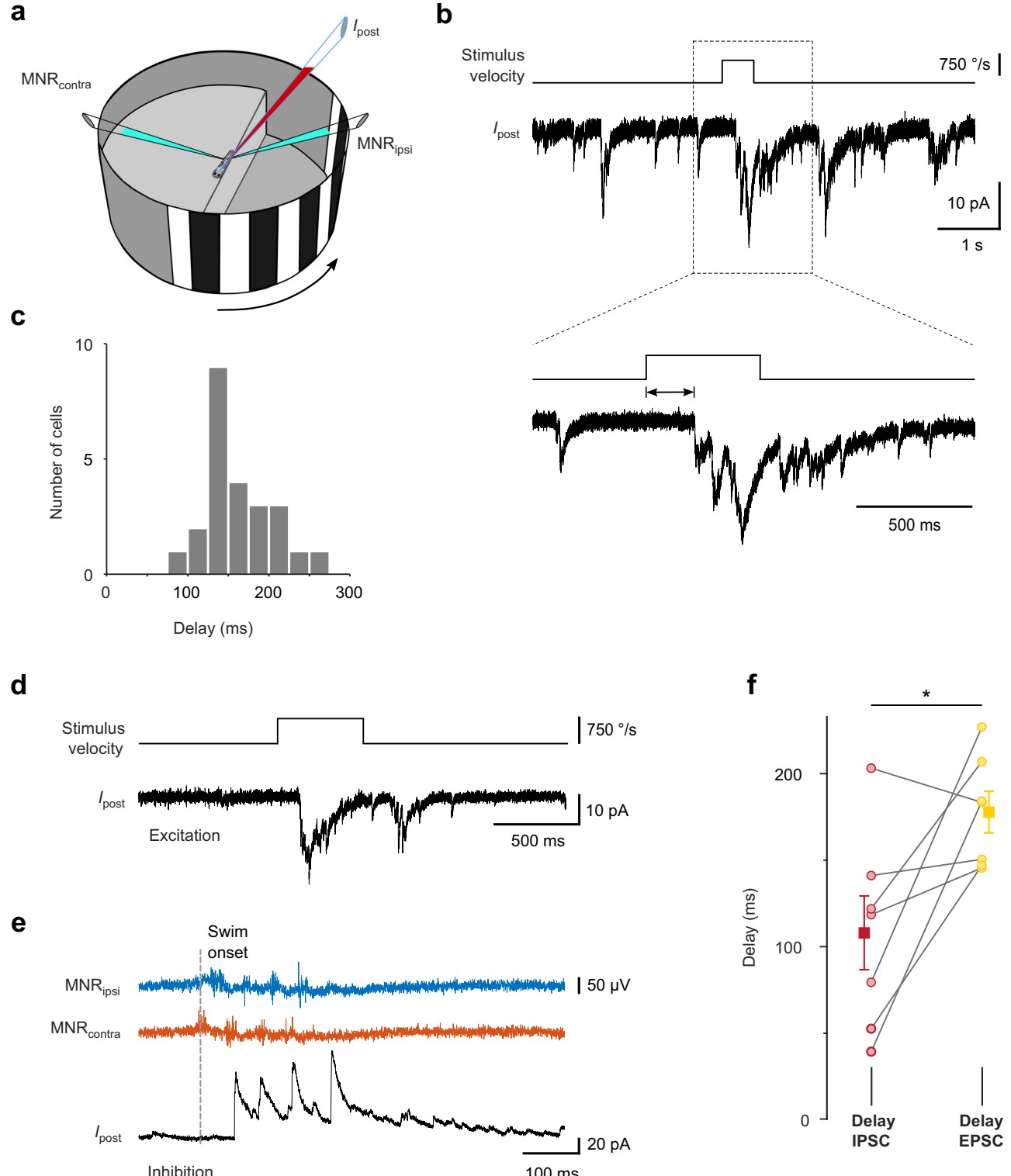

**Fig. 5 | Delay of excitatory input evoked by fast whole-field motion stimuli.**
**a** Experimental configuration. Excitatory currents were recorded in voltage clamp (holding potential −60 mV) from cell contralateral to the stimulus. A stationary grating was shown, which abruptly moved backward, simulating reafferent whole-field visual motion during discrete swim bouts. **b** Example recording of EPSCs ($I_{post}$) during presentation of grating, which moved rapidly for 0.5 s (top trace). EPSC delays were measured from onset of stimulus movement (magnified view, bottom traces). **c** Histogram of delays between stimulus onset and EPSC onset. Individual cell averages from $n = 24$ cells (161.3 ms ± 8.2 ms; mean ± SEM). **d** Example recording of moving grating-evoked EPSC ($V_{hold}$: −60 mV) in a cell in which also

spontaneous swim-related inhibition was measured. **e** Example recording of IPSC ($V_{hold}$: 10 mV) following a spontaneous swim bout. Same cell as in **d**. **f** Pairwise comparison of swim-related IPSC delays (red) and delays of EPSCs from the onset of visual grating motion (yellow), measured in the same cells. Individual cell averages from $n = 7$ cells (circles). Squares and error bars indicate mean ± SEM across cells (EPSC: 178 ms ± 12 ms; IPSC: 108 ms ± 21 ms). Pair-wise difference: 70 ± 25 ms (mean ± SEM, $p = 0.047$, two-sided Wilcoxon signed-rank test). For analysis of EPSC delays, trials in which the visual stimulus evoked swimming were excluded. Source data are provided as a Source Data file.

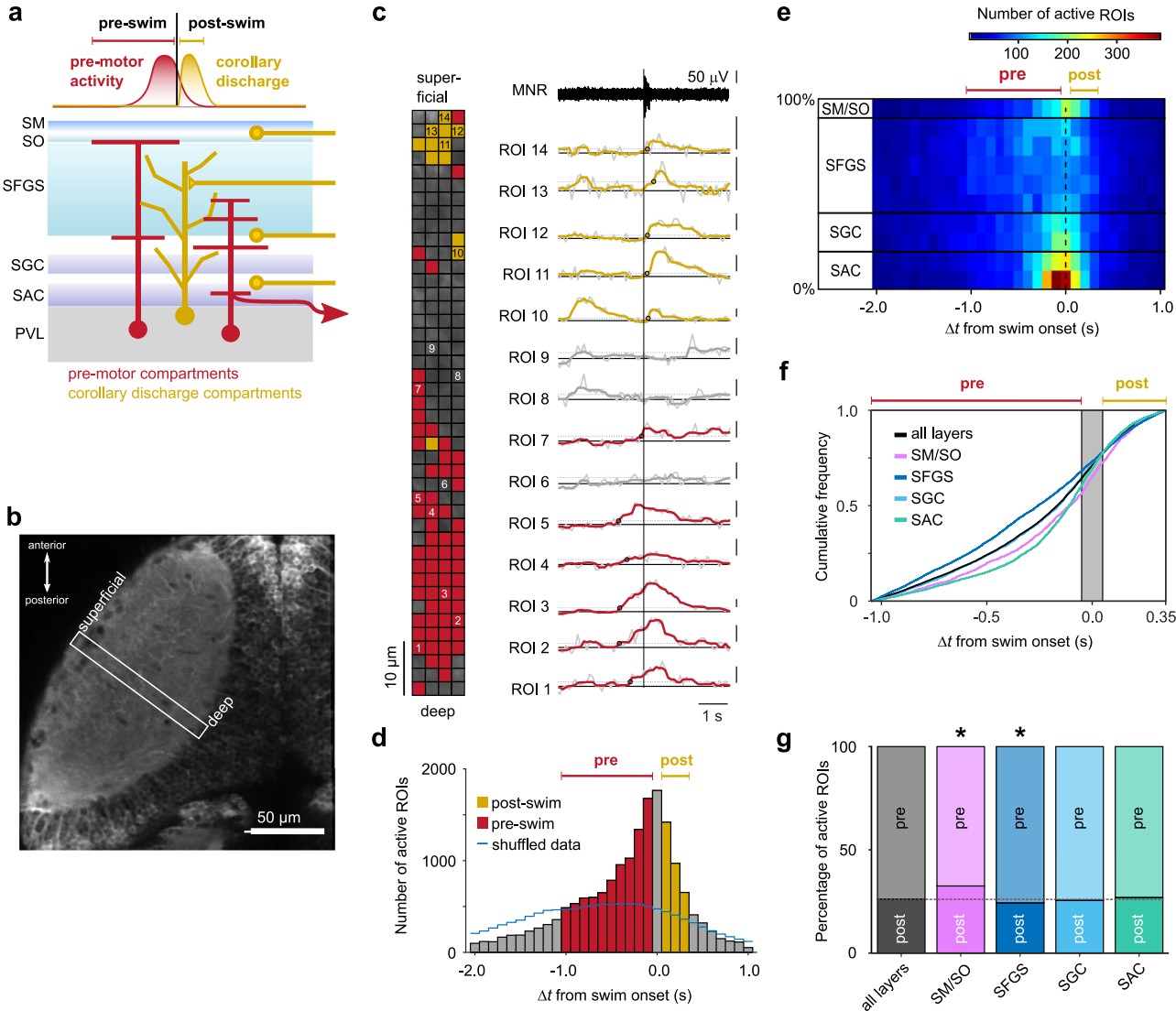

**Fig. 6 | Spatial distribution of swim-related Ca²⁺ signals in the tectal neuropil.**
**a** Schematic of tectal layers, showing interneurons, projection neurons and incoming axons that could mediate premotor (red) or corollary discharge activity (yellow), respectively. **b** Tectal hemisphere in *Tg(elavl3:GCaMP5G)*, dorsal view. Ca²⁺ imaging was performed in a rectangular scan area (white box) covering deep to superficial neuropil during spontaneous fictive swims. Image representative of recordings from 15 larvae. **c** Example of fluorescence transients (right traces) from small ROIs in the neuropil around swim onset (vertical line). Traces are from numbered ROIs in left inset. (Scale bars: 0.4 ΔF/F; raw traces overlaid with median-filtered traces, see "Methods" section). Onset times of Ca²⁺ transients marked by circles. ROIs were considered 'active' with respect to the swim event if a peak was detected whose onset fell in the interval [−1.05 s; 0.35 s] around swim onset (colored traces; see Supplementary Fig. 4). ROIs in inset and ΔF/F traces are colored according to whether their onset was categorized 'pre-swim' (red), 'post-swim' (yellow), or 'inactive' (gray). **d** Histogram of active ROI onsets in relation to swim onset. Active ROIs with onset in the central bin (gray bar at 0 s) were excluded from categorization. Blue line indicates control distribution of active ROIs when fluorescence data was shuffled circularly. Data from 315 spontaneous swims in 15 larvae; total of 16890 active ROIs. **e** Space-time histogram of active ROIs. Active ROIs from

**d** were binned according to location in the neuropil (spatial bin size: 10%). Spatial bins are grouped according to anatomical layers: 0-20%: SAC; 20-40%: SGC; 40-90%: SFGS; 90-100%: SM/SO. **f** Cumulative distributions of active ROIs, pooled over all layers (black trace), and pooled separately for the different neuropil regions indicated in **e**. The cumulative distributions for SM/SO, SFGS and SAC were different from that summed across all layers (SM/SO: $p = 5.7 \times 10^{-7}$; SFGS: $p = 7.5 \times 10^{-29}$; SAC: $9.8 \times 10^{-33}$; SGC: $p = 1.0$. Kolmogorov-Smirnov tests with Bonferroni adjustment of p-values for multiple comparisons). **g** Fractions of active ROIs classified in **d** as 'pre-swim' or 'post-swim', pooled over all layers and pooled separately for the different neuropil regions. In SM/SO, Ca²⁺ transients occurred more frequently in the post-swim interval (32.5%, $p = 2.3 \times 10^{-6}$) compared to the fraction measured across all layers (26.0%, horizontal dashed line). In SFGS, Ca²⁺ transients occurred less frequently in the post-swim interval (24.2%, $p = 1.1 \times 10^{-2}$). In SGC and SAC the fractions were not significantly different from that across all layers ($p = 1.0$ in both cases; all p-values from two-sided binomial tests with Bonferroni adjustment of p-values for multiple comparisons). Events around swim onset (gray bar in **f**) were excluded from categorization. Source data are provided as a Source Data file.

delay was relatively independent of the direction and strength of the fictive swim bout (Fig. 3), this suggests that it has a suppressive effect on tectal output signals during all types of swims.

In other systems, CD signals have been found to encode a "negative image" of the reafferent sensory input due to the animal's self-motion, which is then subtracted from the actual afferent

input[4,15,45]. We observed that the inhibitory charge transfer was correlated with swim power measured on the same side (Fig. 3 and Supplementary Fig. 2). This suggests that in addition to its role of attenuating tectal output, the observed CD signal could encode quantitative information on the expected reafferent visual input during different swim patterns, e.g. when turning in one or the other

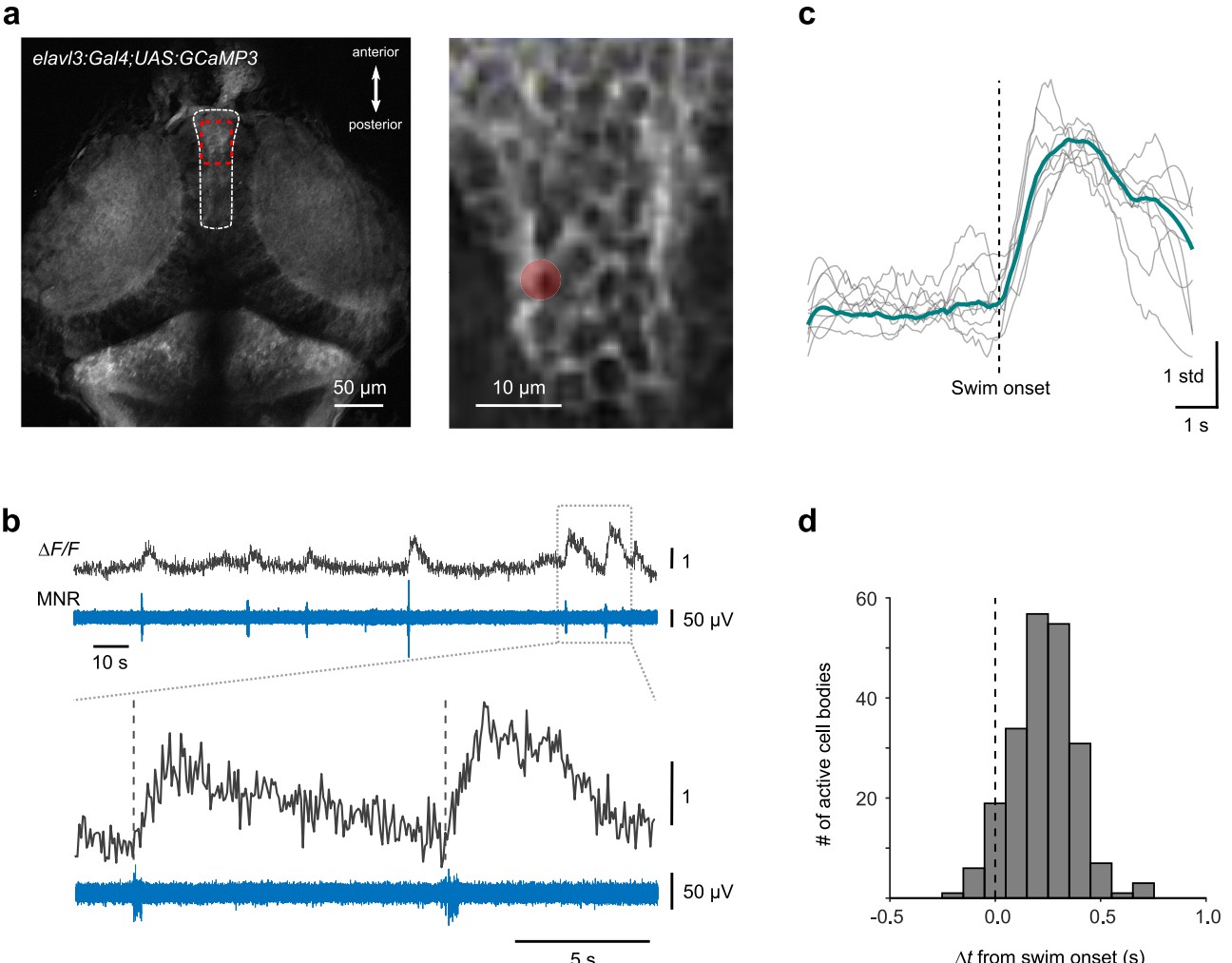

**Fig. 7 | Temporal distribution of swim-related Ca²⁺ signals in the torus longitudinalis (TL). a** TL (dashed white outline) in *Tg(elavl3:Gal4-VP16;UAS:GCaMP3)*, (dorsal view). Rapid Ca²⁺ imaging (8−27 Hz) was performed in a rectangular scan area (red box; magnified view in right inset) during spontaneous fictive swimming in the absence of visual stimulation. Image representative for independent recordings from 5 larvae. **b** Example of fluorescence transients (ΔF/F) from cell body in the TL (red circle marked in (**a**, right inset). ΔF/F transients coincide with spontaneous swim bouts, recorded simultaneously (MNR, bottom). Onset of ΔF/F transients occurs around swim onset (vertical dashed lines in magnified view, lower traces). **c** Swim-triggered average (green trace) of ΔF/F transients from cell shown in **b**. Individual ΔF/F transients (gray traces) were z-scored prior to averaging. **d** Histogram of ΔF/F transient delays from swim onset measured in TL cell bodies. (*n* = 214 cells from 5 larvae). Source data are provided as a Source Data file.

direction in response to appetitive and aversive stimuli[21,22], or at different swim speeds[46]. Thus, the CD signal observed here could in principle be more than a simple signal for saccadic suppression, but may be a component of a predictive signal, distributed across many cell types in visual areas, that represents an 'internal model' of the expected visual input during swimming[47]. In that case, a mismatch between the CD internal model and the actual reafferent input would serve as an error signal that instructs motor learning on slower time scales[29]. Such error signals could be critical in modifying synaptic weights between visual and motor areas in order to adapt the visuomotor transformation to a change in external or internal conditions, such as changes in the surrounding medium, or changes in the skeletomuscular apparatus as the animal grows.

Finally, a transient inhibitory input may also have a facilitating effect[6,48] and sensitize the visual circuitry for subsequent detection of behaviorally relevant, local stimuli, such as prey-like objects. This is because a synchronous, widely distributed inhibitory input will counteract the widespread depolarization of tectal cells expected from the wave of self-motion induced retinal inputs, remove depolarization block after the phasic inhibitory inputs have decayed, and may lower the threshold for generating subsequent, target-directed premotor activity[39,49]. When performing motor sequences to capture prey, zebrafish larvae exhibit the shortest reaction times when the stimulus reappears immediately after a previous swim bout has ended[22]. This behavioral effect could be explained if the stimulus-evoked, retinotopically organized input arrives on a sensitized tectal circuitry, in which activity from previous stimuli and self-motion-induced optic flow has been quenched by a global inhibitory signal. Furthermore, since the inhibitory CD signal effectively forms a negative feedback loop, this system could be prone to oscillatory behavior and contribute to setting the inter-bout interval of swim sequences.

In summary, the CD signal we discovered in the fish optic tectum could serve multiple roles in that (1) it blocks or attenuates tectal output to hindbrain premotor circuits at the time of swimming; (2) it could be the basis for computing an error signal relevant for motor learning if the expected and the actual visual feedback do not match; and (3) it could render tectal circuits more sensitive immediately after a discrete swim bout for accelerated detection of local stimuli and faster behavioral performance. These possibilities fit well with current concepts on the function of various instances of CD signaling

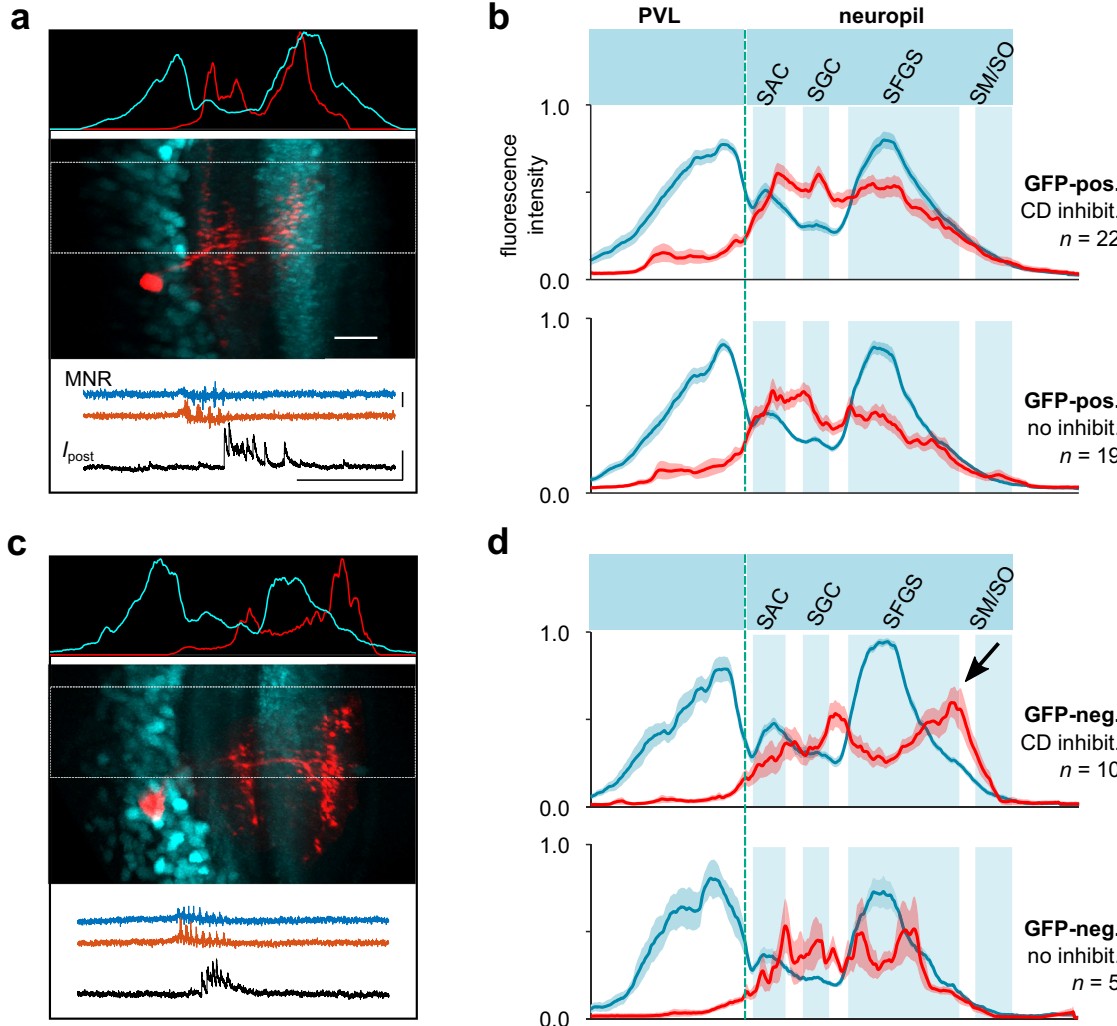

**Fig. 8 | Dendritic profiles of tectal cells with CD synaptic inhibition. a** Dendritic profile of a recorded GFP-positive neuron labeled with sulforhodamine (red) in the *Tg(pou4f1-hsp70l:GFP)* line (cyan). Scale bar 20 μm. Profiles of fluorescence intensity (top inset, peak-scaled) were measured separately in the red and cyan channel of the dual color image (center) across the PVL and neuropil (rectangular area in center image). The recorded neuron received CD inhibitory postsynaptic currents associated with swimming (bottom inset). Scale bars MNR: 50 μV, $I_{post}$ 50 pA, time: 500 ms). **b** Average dendritic profiles of recorded GFP-positive neurons, for cells with motor-related, CD inhibition (top) and without CD inhibition (bottom). Profiles were aligned with respect to the PVL/neuropil boundary (vertical dashed line) and the peak of the SFGS and averaged (cyan traces, mean ± SEM). Profiles of neurons (red) exhibit dendrites mostly in SAC, SGC and central SFGS, but not in more superficial layers. Shaded areas indicate approximate extent of neuropil layers. **c** Same as in **a**, but for a GFP-negative neuron patched in the *Tg(pou4f1-hsp70l:GFP)* line. Note the bistratified dendritic profile, with the upper dendrites in target layers more dorsal to those of GFP-positive neurons. **d** Same as in **b**, but for GFP-negative neurons patched in the *Tg(pou4f1-hsp70l:GFP)* line. Note the peak of dendritic profiles superficial to the SFGS (arrow). Source data are provided as a Source Data file.

identified in numerous species and related to different modalities[2,6], such as vision[11–17,40], audition[8,50], mechanosensory processing[9,10] and electroreception[7,45]. Our findings in the zebrafish now provide clear and direct evidence for a well-defined inhibitory synaptic mechanism in vivo that exerts a suppressive effect on visually evoked spiking activity in a central visual area conserved across all vertebrate phyla. As this signal is robust and occurs in the absence of actual muscle activity, it is not the consequence of proprioceptive feedback but must be the result of internal motor commands. Since the zebrafish model allows for experimental dissection of neural circuits in the intact animal, these results may serve as an entry point for mapping neural pathways and synaptic mechanisms that contribute to internal representations of self-motion in the vertebrate brain. For example, zebrafish exhibit robust optokinetic responses that depend on neuronal populations in the pretectum[51]. It is of interest whether also neurons in the pretectal visual pathway receive an inhibitory CD signal, which may be addressed in future experiments.

What could be the sources of tectal CD signaling? The pathway that evokes the phasic, motor-related inhibitory signal in the tectum must be activated by premotor neural circuitry whose activity triggers the recruitment of motor neurons in the spinal cord. The origin of the CD pathway, from where motor-related signals are sent toward sensory areas is unclear, but we can go backwards and ask by which route motor-related CD signals enter the tectum. Apart from retinal inputs, the tectum receives afferent fibers from several non-sensory areas. Tectal afferents have been described in the larva whose cell bodies are located in the nucleus isthmi in the tegmentum[52,53], the thalamus[54], rostral hypothalamus[55], the raphe nucleus[56,57], pretectum[58,59], the cerebellum[60], and the TL[41]. Most of these regions are unlikely to relay a CD signal to the tectum because they are mainly visually responsive (thalamus, nucleus isthmi)[52–54], or exert state-dependent modulatory influence on tectal processing on slower time scales (hypothalamus, raphe nucleus)[55–57].

By contrast, evidence for the TL as a mediator of motor-related CD signals in the tectum comes from our Ca²⁺ imaging experiments, which

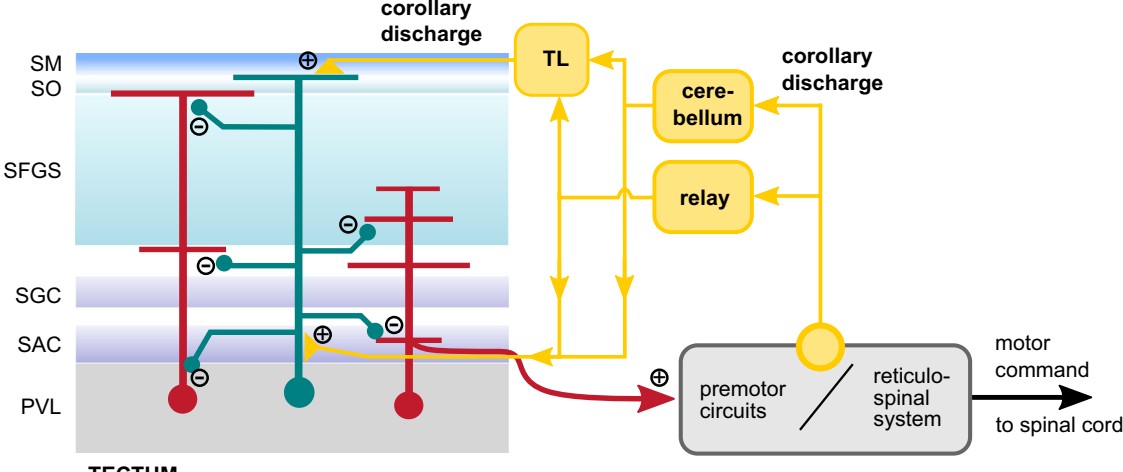

**Fig. 9 | Putative mechanism of CD signaling in the visuomotor pathway.**
Hypothesized organization of inhibitory CD signaling in the tectum. Diverse cell types in the tectum (red cells) receive motor-related inhibitory inputs during swimming. The CD signal is initiated at unknown sites in premotor circuitry controlling tail and eye movements (yellow circle) and is relayed to the tectum, possibly via the cerebellum, or other relay nodes. Because post-swim Ca²⁺ signals cluster in the most superficial layer containing the *stratum marginale* (SM) where axons from TL projection neurons form a narrow input layer, TL is a likely pathway whereby CD signals reach the tectum. This is supported by transient post-swim activity observed in TL neurons (Fig. 7). Similarly, afferents from the cerebellum, or from other relay nodes, which preferentially terminate in deep neuropil layers containing the SAC, could form another input channel for CD signals. Because projection neurons from both the TL and the cerebellum are mainly glutamatergic, this model posits that the afferent CD signal is sign-converted by local inhibitory interneurons (green). As CD likely enters the tectum via superficial TL fibers, this interneuron type is expected to receive excitatory input in the SM/SO and distribute inhibitory signals across several neuropil layers. This could explain how neurons with only deep dendritic branches (Fig. 8) receive phasic inhibitory inputs despite a lack of dendrites in the more superficial layers (red neuron on the right).

showed that transient post-swim activity, whose timing is consistent with that of the motor-related inhibitory input, clustered in compartments in the dorsal SM/SO. It is this layer, where the axons from TL projection neurons form the only source of afferent input and synapse on tectal neurons[41]. Our direct observation of Ca²⁺ transients in many TL neurons immediately starting after the onset of spontaneous swim bouts further corroborates this model. This is in accord with earlier findings that TL neurons in several ray-finned fish species exhibit transient activity during saccadic eye movements[40]. Together, this makes TL projection neurons a likely candidate as a relay for the inhibitory CD signal found in our recordings. The afferent pathways that drive CD activity in TL projection neurons are unclear, but potential sources are cerebellar projection neurons and cells near the cerebellar valvula[40,60,61].

Also, motor-related output from eurydendroid projection neurons[62,63] in the cerebellum is another potential source for CD signals in the tectum. First, axons from some eurydendroid cells densely and extensively innervate deep layers of the tectal neuropil[60]. Second, the activity of eurydendroid cells is correlated with both spontaneous and evoked motor events[64]. The coincidence of eurydendroid activity with swimming is not surprising because their presynaptic excitatory inputs, the parallel fibers of cerebellar granule cells[62,63], also exhibit markedly increased firing rates during brief swim bursts[65]. Therefore, cerebellar output may, either directly or indirectly via the TL, channel motor-related information into the tectal neuropil. It should be noted, however, that both cerebellar eurydendroid cells and most or all TL granule cells are glutamatergic[40,41,63], so this possible CD-pathway requires at least one additional sign-converting relay neuron. Several types of inhibitory interneurons have been identified in the tectum that could serve this role[26,66].

In summary, the evidence from our data combined with results from earlier studies on the anatomy and function of cerebellar projections favors a model of a feedback loop sending a CD signal from the premotor circuitry in the midbrain and hindbrain, processed by the cerebellum and relayed via the TL, and possibly other relay nodes, to the tectum (Fig. 9).

In mammals, the emergence of thalamo-cortical visual circuits in addition to the evolutionarily conserved visual pathways in the brainstem provides for a fundamental extension of visual processing power. Not surprisingly, motor-related modulation is ubiquitous in cortical processing[47]. Indeed, evidence for perisaccadic modulation of neuronal activity during eye movements is well established in several cortical areas[12-14,67,68]. Here, CD signals are considered a critical factor underlying perceptual saccadic suppression and enhancement[3,48], which enables perceptual continuity during self-generated eye movements. It is of note that CD signals implicated in cortical processing are likely mediated by the midbrain superior colliculus[3,67], thus forming a functional bridge between subcortical and cortical visual processing. But more than just being the source of CD, superior colliculus neurons themselves are a target of motor-related modulation as their responses are altered during saccadic eye movements in cats[69] and primates[11,70], and also during locomotion in the rodent superior colliculus[71]. This is similar to the suppression of activity we observed in visually responsive interneurons as well as output neurons in the optic tectum that project to motor areas. In the mammalian superior colliculus, an intracollicular inhibitory circuit motif could in principle mediate CD-like inhibition in visual neurons in more superficial layers[72], but also purely retinal mechanisms could contribute to it[73]. The mammalian superior colliculus and its homologous region, the optic tectum in non-mammalian vertebrates, share fundamental properties[74,75]. Our results suggest that a temporally robust inhibitory synaptic input driven by an extratectal circuit mediates this suppression in the optic tectum in zebrafish during saccade-like locomotion. In the mammalian superior colliculus, recordings of saccade-related synaptic inputs may provide cues for potentially similar synaptic mechanisms.

## Methods

### Zebrafish

Zebrafish larvae (*Danio rerio*) were raised at 28.5 °C in embryo medium in a 14 h/10 h light/dark cycle. Experiments were performed on zebrafish larvae 5–8 days post-fertilization, before sexual differentiation. The following transgenic lines were used: For electrophysiological

recordings and immunohistochemical analysis, *Tg(pou4f1-hsp70l:GFP)*[31,32], in which GFP is expressed in retinal ganglion cells and a subset of tectal periventricular neurons. For Ca²⁺ imaging, *Tg(elavl3:GCaMP5G)*[38] or *Tg(elavl3.1:Gal4-VP16;UAS:GCaMP3)*[26] were used in the *nacre* background[76], in which the genetically encoded Ca²⁺ sensors GCaMP5G or GCaMP3 were expressed pan-neuronally[26,38,77]. Animal husbandry and experimental procedures were performed following the guidelines of the German animal welfare law and approved by the local authorities (Regierungspräsidien Karlsruhe and Freiburg).

### Electrophysiology and visual stimulation

**Preparation and solutions.** For electrophysiological recordings, zebrafish larvae were anesthetized using 0.02% MS-222 (tricaine methanesulfonate, Sigma Aldrich) and then immobilized by incubation in α-bungarotoxin (1.0 mg/ml dissolved in embryo medium, Tocris). The larva was transferred to a custom-made cylindrical recording chamber (5.0 cm outer diameter, 2.5 cm height). It was mounted in an upright position at mid-height of the chamber on a Sylgard-shelf that covered 50% of the bottom to leave an unobstructed view for visual stimulation (see Fig. 3a). The chamber was filled with extracellular solution containing (in mM): NaCl (134), KCl (2.9), CaCl₂ (2.1), MgCl₂ (1.2), HEPES (10), and glucose (10), pH 7.8, 290 mmol/kg. To approach tectal cells for whole-cell recordings, a small opening was made in the skin above the optic tectum using an etched tungsten needle. The preparation was then moved to a custom-built two-photon laser scanning microscope equipped with infrared transmission optics to enable positioning of recording electrodes for electrophysiological experiments. Targeted patch clamp recordings of GFP-expressing cells were established using a water immersion objective (Zeiss 20x/1.0 NA) and two-photon laser excitation at 940 nm. Emission light was spectrally separated using two bandpass filters (green: HQ 515/30 nm; red: HQ 610/75 nm, AHF, Germany).

**Recordings.** To record fictive motor activity, two suction electrodes (borosilicate glass pipettes with heat-polished tips; open tip diameter ~30 μm) were filled with extracellular solution and positioned on intersegmental boundaries in the midrange of the intact tail, one on each side, and gentle suction was applied[28]. Bilateral motor nerve activity was recorded in current clamp mode, low-pass filtered at 3 kHz. Once spontaneous motor nerve activity was detected, a whole-cell recording from a tectal cell was established under visual control using the two-photon fluorescence optics of the microscope by which fluorescence of GFP-labeled cells and the patch pipette containing sulforhodamine could be monitored in real time[30]. Patch pipettes were pulled from borosilicate glass and filled with intracellular solution containing (in mM): K-Gluconate (125), HEPES (10), EGTA (10), MgCl₂ (2.5), ATP-Na₂ (4), and GTP (0.3), Sulforhodamine-B (360 μM, Invitrogen), pH 7.3, 290 mmol/kg. Open tip resistance was 8–14 MΩ. In some recordings, K-Gluconate was replaced with Cs-Gluconate (120 mM) to minimize leak potassium currents during voltage clamp. Inhibitory synaptic currents were measured at a holding potential of 0–20 mV. Whole-cell current and voltage signals were recorded using a Multiclamp 700B amplifier and filtered at 3 kHz. The simultaneous bilateral motor nerve recordings and the single-cell patch clamp recording were sampled at 10 kHz on the same computer using a PCIe-6251 board and a custom-written script in LabVIEW (2013–2021, National Instruments, USA).

**Visual stimulation.** Visual stimuli were projected on a diffusive screen (Ecolor#216, Rosco) attached to the side wall of the recording chamber, using a microprojector (Optoma PK102, or Kodak Luma 75). The projection area covered ~120° (azimuth) and 55° (elevation) of the visual field of the stimulated eye, without obstruction from the sylgard shelf or distortions from water-air interfaces (compare Fig. 3a). To evoke target-directed, prey tracking-like swims, a small white rectangle moved horizontally on a dark background (target size ≤6° height, ≤12° width)[22]. To evoke avoidance swims away from the target, a large white rectangle (≥12° height, ≥24° width) moved on a dark background or an expanding dark disc was shown on a white background ('looming object'). Moving targets appeared in the caudal visual field, moved forwards and then backwards at a speed of 70°/s, covering ~100° of the horizontal visual field (from −110° to −10°, where 0° refers to the heading direction of the larva). Expanding discs appeared in the center of the visual field of the stimulated eye and expanded at a rate of 70−90°/s. For measuring the latency between the onset of whole-field visual motion and excitatory input to tectal neurons (Fig. 5), a stationary grating was presented to the eye contralateral to the recorded neuron. The grating then moved for 0.5 s in a front-to-back direction at a speed of 750°/s. The onset of grating motion was detected with a photodiode placed in the visual stimulation pathway and sampled at 10 kHz on the same A/D-board as the single-cell recording to precisely measure latencies between stimulus onset and synaptic responses. Visual stimuli were programmed using Python based OpenGL Vision-Egg Software[78].

**Analysis of electrophysiological data.** Electrophysiological data were analyzed offline using scripts programmed in LabVIEW (2013–2021, National Instruments, USA) and MATLAB (2021a, The MathWorks, USA). Peripheral motor nerve recordings were rectified by calculating the standard deviation in a 10-ms moving window, low-pass filtered at 500 Hz and median-subtracted. Fictive swim bouts were identified in the smoothed record using a threshold criterion of typically 4 standard deviations (SD) in each trace. The number of peaks exceeding the threshold ('bursts') and their location was measured in each trace. The average peak distance was taken as interburst interval. The difference between last and first peak location plus one interburst interval was taken as bout duration. The location of the first peak in the two traces, whichever came first, was taken as swim onset. Unilateral swim power ($Power_{ipsi}$, $Power_{contra}$) in a bout was measured in each of the two recordings as the area under the smoothed curve within the calculated bout duration. The summed swim power ($Power_{ipsi}$ + $Power_{contra}$) was taken as a measure of swim strength. To infer the direction of the swim, a direction index was calculated as ($Power_{ipsi}$ − $Power_{contra}$)/($Power_{ipsi}$ + $Power_{contra}$), with positive values indicating a swim directed towards the stimulus, ipsilateral to the stimulus position.

To quantify motor-related inhibitory synaptic currents in tectal neurons, the voltage clamp recording was baseline-subtracted using its mean in a 0.2 s-interval before swim onset. The charge transfer was calculated by integrating the recorded current trace in a window starting 50 ms after swim onset for 250 ms. When analyzing spontaneous swims, cells without discernible phasic IPSCs clustered around 0 pC charge transfer (Fig. 2c), and were taken as receiving no motor-related inhibition. Cells that exhibited sharp rising, phasic IPSCs typically had charge transfer ≥0.8 pC. IPSC onset was measured as the time of crossing a threshold of typically 3 SD in the current trace, from which the IPSC delay relative to swim onset was measured. The delay between onset of whole-field image motion on the retina and excitatory inputs in tectal cells (Fig. 5) was determined correspondingly.

To measure motor-related voltage signals, swim-triggered voltage traces were base-line subtracted and averaged for each cell, from which an average across cells and SEM-traces were calculated (Fig. 1e). Visually evoked spiking activity was quantified by identifying spikes using a rate of rise criterion ≥5 mV/ms to determine spike times (Fig. 4). Assuming that the spike output from the recorded population of cells converges on a single premotor target, a population spike count was calculated in bins of 100-ms (Fig. 4b, f). For each cell, the average number of spikes across trials was calculated for each bin and summed across cells. Spike frequency was calculated as the average spike count per 100 ms in each bin for each cell (Fig. 4e). For determining the time-resolved spike probability ($P_{spike}$, Supplementary Fig. 3a) for cells that

receive motor-related inhibition vs. those that do not, the summed spike count histogram was determined for each group separately and divided by the total number of spikes in the interval [−1.5 s; 2.0 s] around swim onset.

**Morphological analysis.** Following electrophysiological recording, a dual color image stack of the recorded neuron and the surrounding GFP-labeled structures in the *Tg(pou4f1-hsp70l:GFP)* background was acquired. GFP-fluorescence bleed-through was subtracted from the red channel sub-stack. Image stacks were cropped where no structures of the recorded neuron were detectable to remove background fluorescence from spilled indicator and to better bring out the position of GFP-positive tectal layers near the recorded neuron. To analyze dendritic morphology, image stacks were rotated to obtain a side view of the recorded neuron. Line profiles were taken from each color channel separately of the maximum intensity projection of the rotated stack and peak-scaled. The red and green channel line profiles from all recorded cells were then aligned with respect to the PVL/neuropil boundary and the center of the SFGS fluorescence, and averaged (Fig. 8).

## Ca²⁺ imaging

**Data acquisition.** For $Ca^{2+}$ imaging of swim-related activity in the tectal neuropil, zebrafish larvae were anesthetized using 0.02% MS-222 and then immobilized by incubation in α-bungarotoxin as described above. The larva was mounted, dorsal side facing upward, in a cylindrical recording chamber filled with embryo medium. $Ca^{2+}$ imaging was performed using a multiphoton laser scanning microscope (Olympus FluoView FV1000, Software Version 4.2a), equipped with a water immersion objective (Olympus 20x/1.0 NA) and coupled with a Ti-Sapphire laser at an excitation wavelength of 920 nm. Emission light was filtered with a bandpass (515–560 nm). During imaging of tectal activity, spontaneous fictive motor activity was recorded unilaterally from the tail using a suction electrode as described above.

Spontaneous $Ca^{2+}$ signals in the tectal neuropil were recorded in the dark, in multiple sweeps per larva, lasting 300 s each (Fig. 6). At a depth of 30 μm from the dorsal most point of the tectum, a rectangular scan region (-100 μm × 10 μm) covering all layers of the tectal neuropil was selected. Imaging was performed at a spatial resolution of typically 320 by 32 pixels, at a frame rate of 9–11 Hz. The fast scan direction was along the long axis of the rectangle. Spontaneous motor nerve activity was acquired at 10 kHz using LabVIEW (2013–2021, National Instruments, USA). $Ca^{2+}$ imaging and motor nerve recordings were synchronized using a trigger programmed in Python (Version 2.7).

For imaging cell bodies in the TL during spontaneous fictive swimming (Fig. 7), a rectangular region covering a central segment of the TL was scanned at a depth -30–50 μm from the dorsal skin at frame rates of 8–27 Hz.

**Analysis of Ca²⁺ imaging data.** $Ca^{2+}$ imaging and MNR data were analyzed using MATLAB scripts (2021a, The MathWorks, USA). Fictive swimming activity was automatically detected using the *findpeaks* function on the cubed and rectified MNR signal. Analysis was performed on isolated swim bouts, separated from preceding and following swim events by at least 3 seconds. Swims occurring within 10 s of turning on the laser were excluded, as well as motor nerve activity lasting less than 150 ms or more than 500 ms. The first peak of a selected swim bout was taken as swim onset.

To analyze localized $Ca^{2+}$ signals in the tectal neuropil, the scanned rectangular region was subdivided into a regular grid of regions-of-interest (ROIs). Each ROI covered a square of approx. 2.8 μm × 2.8 μm of the scanned region, with shared edges between adjacent ROIs. For each individual ROI, the raw fluorescence time course was extracted and detrended using a quadratic fit. Fluorescence time course in each ROI was then expressed as $\Delta F/F = (F(t) − F_O) / F_O$, where $F(t)$ is the fluorescent intensity at a given time point and $F_O$ is the mean of the lowest 50 $F(t)$ values.

After alignment with the MNR recording, for each detected swim a 9 s time window around swim onset was cropped out from the $\Delta F/F$ traces for each ROI. The traces were smoothed using a sliding median window of length 5 frames and then subjected to an automatic $Ca^{2+}$ transient detection method. For each cropped trace, a baseline was determined as mean of the lowest 60% of intensity values. A standard deviation (SD) was calculated from the 33% lowest values. Using the *findpeaks* function, peaks in the $\Delta F/F$-trace that exceeded the baseline by > 6 SD and had a width of at least 4 sample points were detected as a potential $Ca^{2+}$ signal. Stepping back in time from the peak, the interpolated time point at which the $\Delta F/F$ trace crossed the 3-SD-level above baseline was taken as the onset of the $Ca^{2+}$ signal. If multiple peaks were detected in a cropped trace, only the one closest to the swim onset was taken for further analysis. $Ca^{2+}$ transients were considered to reflect burst-like spiking activity potentially related to the swim bout (Fig. 6d-g) if the following criteria were met: (1) the rising phase, taken as the time between the $Ca^{2+}$ transient onset and its peak, was <1.5 s (2) the onset of the transient occurred in an interval [−1.05 s; 0.35 s] around swim onset. The difference between $Ca^{2+}$ transient onset and swim onset ($\Delta t$) was used to classify $Ca^{2+}$ transients as 'pre-swim' ($−1.05\,s \leq \Delta t \leq −0.05\,s$) or 'post-swim' ($0.05\,s \leq \Delta t \leq 0.35\,s$).

For shuffled control data (Fig. 6d, Supplementary Fig. 4), each recorded fluorescence sweep was circularly shifted by a random time interval chosen from a uniform distribution. The analysis was then done in the same way as for the original data. The shuffling process was repeated 100 times, from which the histogram was calculated (blue line in Fig. 6d). Cumulative distributions and bar graphs representing 'pre-swim' and 'post-swim' $Ca^{2+}$ transients (Supplementary Fig. 4c, d) were generated analogously to those for the original data.

## Immunohistochemistry

**Immunohistochemistry.** Larval zebrafish (5 dpf) of *Tg(pou4f1-hsp70l:GFP)* were anesthetized and transferred to 4% paraformaldehyde (PFA) in phosphate buffer saline (PBS) overnight at 4 °C. Subsequently, the tissue was washed in PBS. Non-specific binding sites were blocked by incubation of the tissue in blocking solution (5% normal goat serum (Sigma, G6767), 1% blocking reagent (Roche, 11096176001), 1% bovine serum albumin (Sigma, A3059) in PBS containing 0.1% Tween 20, 1% DMSO) for 2 h at room temperature. Whole brains were incubated with anti-GFP (chicken anti-GFP, 1:500, Invitrogen A10262) and anti-GABA$_A$ (mouse anti-GABA$_A$ receptor β2,3 chain, 1:100, Merck MAB341). To visualize specific binding of the primary antibodies, the tissue was rinsed with buffer and incubated in secondary antibodies overnight at 4 °C. GFP (anti-chicken 488 (1:1000), Invitrogen A11039) and GABA$_A$ (anti-mouse 546 (1:1000), Invitrogen A11003) were detected with appropriate secondary antibodies conjugated to Alexa Fluor dyes. Finally, the tissue was rinsed in PBS and mounted in mounting medium (80% glycerol, 1% agarose in PBS) for further analysis.

**Analysis.** Whole mount imaging of the immunolabelled tectal hemispheres was acquired using a laser scanning confocal microscope (FluoView 1000, Olympus) with a water-immersion objective (Olympus 20x/1.0 NA). Green fluorescence was excited at 488 nm, red fluorescence was excited at 561 nm. Images were acquired with a spatial resolution of 1024×1024 pixels. Confocal images were processed in ImageJ (Version 1.53, using the FIJI distribution package). Rectangular ROIs (width: 40 μm, length: 55 to 120 μm) were defined ranging from the PVL/neuropil boundary to the dorsal boundary of the tectal neuropil, at comparable depths across all samples. A median-filtered image of the ROI was subtracted from the raw image. The processed ROI was thresholded using maximum entropy thresholding. For the

quantification of stained puncta, the Analyze Particle command was used. Particles >8 pixels and a circularity of >0.3 where counted as immuno-labeled puncta. To account for differences in neuropil extent across samples, neuropil depth was normalized to 100%, binned in 10% steps. For the 10 subregions, the proportion of puncta in every bin was calculated by dividing the number of puncta in each spatial bin by the sum of puncta across all bins.

## Statistics

Graphical presentation of data and statistical tests were done using MATLAB (2021a, The MathWorks, USA). All population data are expressed as mean ± standard error of the mean (SEM), if not stated otherwise. For statistical tests, a significance level of $\alpha = 0.05$ was chosen, and where applicable, $p$-values were adjusted using Bonferroni's method or Tukey-Kramer's method for multiple comparisons. Box and whisker plots indicate the median (center line), and the upper and the lower quartiles (box). Whiskers mark the upper and the lower limit of the data range, up to a maximum of 1.5 times the interquartile range (that is, the width of the box) from the upper and lower edge of the box, respectively.

## Reporting summary

Further information on research design is available in the Nature Portfolio Reporting Summary linked to this article.

## Data availability

The source data for all figures and supplementary figures in this work are provided with this paper. Raw data are available upon request to the corresponding author. Source data are provided with this paper.

## Code availability

Custom code used in this study for detecting $Ca^{2+}$ transients in time series data from two-photon fluorescence microscopy is available at https://github.com/bollmannlab/Corollary_Discharge/.

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

## Acknowledgements

We thank Margit Böhler and Harald Noeske for expert technical help, and Sabine Götter for excellent help with fish husbandry. We thank Andrew Straw for comments on an earlier version of the manuscript and Wolfgang Driever, Shagnik Chakraborty and Nils Brehm for

helpful discussions. JHB thanks Winfried Denk for support. We thank Hitoshi Okamoto and the National Bioresource Project of Japan for providing the *Tg(pou4f1-hsp70l:GFP)* line. This work was performed with support from the Max Planck Society and the German Research Association (DFG; Project-Nr. 357057764, 398417145, 453632629 and 357057560, JHB).

## Author contributions

M.A.A., K.L., S.J.P., C.A.T., and J.H.B. designed the experiments. M.A.A., K.L., S.J.P., and C.A.T. performed the experiments. M.A.A., K.L., S.J.P., C.A.T., and J.H.B. analyzed the data. J.H.B. wrote the manuscript with help from M.A.A. and K.L. and feedback from C.A.T. and S.J.P.

## Funding

## Competing interests

The authors declare no competing interests.
