## [Peer Review File · Nature Communications]

A Synaptic Corollary Discharge Signal Suppresses Midbrain Visual Processing During Saccade-Like LocomotionREVIEWER COMMENTS

Reviewer #1 (Remarks to the Author):

In this manuscript, the authors report a corollary discharge signal in the optic tectum related to movement. They report that this signal is inhibitory, directly correlated with power of the motor output, and acts to suppress visually evoked spiking in tectal cells. This has not previously been demonstrated in the zebrafish visual system, making this work an interesting finding relevant to the field of sensory-motor integration. However I do have a few concerns.

- In Figures 2 and 3, the authors report an inhibitory current related to motor output. How do the authors know that the current is inhibitory? Were glutamate receptor blockers used in the bath? Was the current sensitive to GABA A antagonists? Did the authors check the reversal potential of the current? The authors also do not report any kinetics of the current that would be consistent with GABAA-mediated chloride currents.
- In Fig 2E there are 19 cells with dV in the range of roughly 0-3, and 2 cells with a DV 3 times larger than this. The authors might want to consider if these latter points should be considered outliers. It looks by eye like there would still be a significant correlation, but it would have a much steeper slope.
- In Fig 2A it would be more helpful to use ipsi and contra labels rather than left and right.
- The illustration of the data in Fig. 3H is difficult to read. Maybe instead / as well the authors could plot stacked 2D curves, which could make trends in the data clearer.
- In Fig. 4A, what were the relative numbers of cells showing the different spiking responses? Why is the probability of spiking shown in Fig. 4G instead of the total number of spikes as shown before? For 4G-4I it would be helpful to demonstrate that these effects are statistically significant. Also in Fig 4 it would be interesting to see the spatial distribution of these cells in the tectum with their degree of inhibition labeled for each cell (e.g. using a color scale).
- The authors say that this inhibitory motor related signal is sufficient to shunt visual evoked excitatory inputs from the timing of the two events. However, there seems to be heterogeneity in tectal cells showing the inhibitory current. Did the cells recorded in Fig. 5 also show the inhibitory motor-related current? How do we know that the excitatory input timing is not heterogenous as well? Given this

heterogeneity in responses, a direct way to show the shunting would be to record excitatory and inhibitory currents in the same cells and analyze trials with or without motor output.

- In Fig 6C the temporal resolution of the measurements is low (100ms) relative to the timing differences that are being inferred. Can the authors provide evidence that these inferred timing differences are robust?

- In Fig 6F which if any of these distributions are statistically different (e.g. using a Kolmogorov-Smirnov test)?

- Fig 7: Why do only some cells receive CD inhibition even though GABA-A labeling is quite diffuse?

- Corollary discharge has been previously shown in the optic tectum of other animals including goldfish, fly, crayfish and mormyrid fish. It would be helpful if the Discussion could include how this study fits into this context of previous findings and what it adds.

- At lines 324-325 the authors say that the CD signal was robust to bout strength. However that's not what they show in Fig 3, and indeed it makes more sense for this dependence to exist.

Reviewer #2 (Remarks to the Author):

This paper investigates a fundamental issue in sensory neuroscience – how the brain handles sensory inputs that originate from self-motion rather than events in the external world. It has long been suggested that signals mirroring motor commands might project to sensory systems to suppress activity generated during self-motion and such a mechanism has been studied in several contexts, including mechanosensory inputs from the lateral line system of fish that are activated during swimming and visual inputs in *Drosophila* activated during flight. This paper now investigates the visual system of larval zebrafish and demonstrates that the output from the optic tectum is transiently suppressed during “saccade-like” swimming bouts due to activation of local inhibitory neurons. As noted in the Introduction, this observation mirrors the known suppression of visual activity in the superficial layers of the superior colliculus of rodents and primates during saccadic eye movements. This is nonetheless a noteworthy result because it is, I believe, the first direct demonstration of suppression of visual processing by an efference copy signal in zebrafish. This will in turn open up important possibilities for a dissection of the overall sensory-motor circuits controlling the interaction between motor activity and

visual processing, for instance by whole brain imaging at single neuron resolution. The model provided in Fig. 8 provides an overview of the directions in which this research might take.

The work is carried out very carefully and methodically, carefully documenting various properties of the inhibition, such as the delay to its initiation. The consequence of this inhibition on visually-evoked responses is also demonstrated. The data analysis is appropriate and the interpretations are tightly linked to the results. I don't believe that further experiments are required to support the major conclusions. The paper is written clearly and provides the methodological detail for others to repeat the experiments. Together, this work is carried out to a very high standard, providing strong support for the major conclusion that visual input is suppressed during self-motion by inhibition tightly coupled to motor activity.

Major criticisms

Fig. 7 and associated text is essentially a negative result that does not advance the paper's conclusions. I think this investigation of GABA receptor distribution in the tectum could be cut without loss.

The plot in Fig 3H is hard to see and interpret. I suggest a plot in 2D (x-axis charge, y-axis power) with ipsilateral and contralateral points in different colours.

Reviewer #3 (Remarks to the Author):

The paper by Ali et al. 'A Synaptic Corollary Discharge Signal Suppresses Midbrain Visual Processing During Saccade-Like Locomotion' describes a midbrain (tectal) mechanism that seems to serve the suppression of visual self-motion signals. Specifically, the authors obtain intracellular (whole-cell) recordings in the tectum of zebrafish larvae. When triggering to the fish self-motion in fictive swimming they observe an inhibitory signal. Subsequently, the authors show that such inhibition is appropriately timed to suppress visual self-motion-inputs; they also show that this signal is present in visually directed swimming. The authors show that the inhibitory signal enters the visual superficial layers of the zebrafish tectum.

My views on this paper are the following. There is already a lot of evidence from primate and other preparations on the neural mechanisms (Wurtz, Thiele etc) of saccadic suppression, which is appropriately cited by the authors. Despite such prior evidence, I still think that this paper is a big step forward. The reason for this assessment is first, the deep mechanistic analysis by the authors (intracellular recordings and high-resolution imaging), which brings us closer to circuit understanding of self-motion suppression. Second, the quite comprehensive investigation of self-motion-related signals by the authors is decisive, because it convinces the reader that there is really a corollary discharge at

work in this preparation. Demonstrating this is an important step for the zebrafish field, because without such evidence we cannot assume that visual self-motion-sppression is at work in this preparation.

I have only one major point

1) The most interesting question is clearly about the origin of the inhibitory signal. The authors discuss to distinct possible sources of this signal (torus longitudinalis projections or the cerebellum. It is not clear to me, why these two candidate mechanisms could not be differentiated by a quick and dirty experiment such as chopping off the cerebellum. I thought the feasibility of such ad hoc interventions is a major advantage of the zebrafish prep.

Minor

Figure 6 C: showing more nonresponses such as ROI4 would make the figure more interesting and convincing.

Point-by-point response to reviewers' comments

We thank the reviewers for their constructive feedback, which we have now addressed with additional experiments, improved and extended analyses, and substantial revisions of the manuscript.

Reviewer #1 (Remarks to the Author):

In this manuscript, the authors report a corollary discharge signal in the optic tectum related to movement. They report that this signal is inhibitory, directly correlated with power of the motor output, and acts to suppress visually evoked spiking in tectal cells. This has not previously been demonstrated in the zebrafish visual system, making this work an interesting finding relevant to the field of sensory-motor integration. However I do have a few concerns.

We thank the reviewer for his/her positive assessment.

- In Figures 2 and 3, the authors report an inhibitory current related to motor output. How do the authors know that the current is inhibitory? Were glutamate receptor blockers used in the bath? Was the current sensitive to GABA A antagonists? Did the authors check the reversal potential of the current? The authors also do not report any kinetics of the current that would be consistent with GABAA-mediated chloride currents.

Thank you for pointing out this important aspect. We have now addressed this question with additional experiments and further analysis. A pharmacological characterization of the motor-related synaptic current is difficult because blocking glutamate receptors will abolish all activity in the tectum and beyond, and application of GABA-A antagonists tends to evoke seizure-like, epileptiform activity (e.g. Baraban et al., *Neuroscience* 131, p759-68 (2005); Bandara et al, *Neurotoxicology* 76, p220-34 (2019)). Therefore, we followed the reviewer's suggestion and performed new experiments in which we measured the current (charge)-voltage-relationship of swim-related synaptic currents (Supplementary Fig. 1a,b). The negative reversal potential at around -50 mV is consistent with a synaptically mediated chloride conductance. We also quantified the time course of unitary, motor-related synaptic currents in which the kinetics of these currents could be measured. This is now mentioned in the manuscript (page 6, first para). The rise time and the decay time constant were consistent with a fast GABA-A mediated synaptic current, which is now presented in Supplementary Fig. 1c,d.

- In Fig 2E there are 19 cells with dV in the range of roughly 0-3, and 2 cells with a DV 3 times larger than this. The authors might want to consider if these latter points should be considered outliers. It looks by eye like there would still be a significant correlation, but it would have a much steeper slope.

To test whether the correlation between synaptic input strength (IPSC-charge) and the observed transient hyperpolarization (ΔV) in the same cells depends critically on the two most extreme data points, we repeated the correlation analysis with the remaining data points only. The

data exhibited still a significant correlation. The corresponding plot is shown in the new Supplementary Fig. 1e.

- In Fig 2A it would be more helpful to use ipsi and contra labels rather than left and right.

Thank you for spotting the inconsistent labeling, which has been changed in the revised version.

- The illustration of the data in Fig. 3H is difficult to read. Maybe instead / as well the authors could plot stacked 2D curves, which could make trends in the data clearer.

We agree. For reasons of space in Fig. 3, we added a new Supplementary Fig. 2, in which the data of Fig. 3h are now shown as suggested by reviewer 1 (and reviewer 2). In addition, we calculated the Spearman correlation coefficients for the two two-dimensional datasets separately, which yielded the same result as in Fig. 3h: IPSC-charge is significantly correlated with swim-power on the same side of the recorded neuron, but not with that on the opposite side.

- In Fig. 4A, what were the relative numbers of cells showing the different spiking responses? Why is the probability of spiking shown in Fig. 4G instead of the total number of spikes as shown before? For 4G-4I it would be helpful to demonstrate that these effects are statistically significant. Also in Fig 4 it would be interesting to see the spatial distribution of these cells in the tectum with their degree of inhibition labeled for each cell (e.g. using a color scale).

Thank you for pointing this out. Fig. 4A shows select examples to illustrate the various effects a measured swim event could have on membrane voltage in the recorded neurons. As the time course of the spiking response from trial to trial was often variable within the same cell, we did not attempt to group the cells based on spike timing during looming stimuli. Instead, we grouped the neurons based on whether or not they received appreciable swim-related inhibition (>0.8 pC, $n = 10$ cells) or not (<0.8 pC, $n = 11$ cells), measured in the same cells using voltage clamp. What we could show this way is that cells with synaptic inhibition showed a consistent transient hyperpolarization during looming evoked depolarizations (Fig. 4c,h and i), and that this hyperpolarization led to a significant reduction in spike count after swim onset, when summed across the population of cells with synaptic inhibition, but not in that without inhibition (Fig. 4g). As suggested, we now present the differences in spiking between the two groups as summed spike count histograms (Fig. 4g), which is indeed more consistent with the presentation in panel 4f. We moved the graph showing this effect in terms of spiking probability to the new Supplementary Fig. 3a, which we originally chose because the probability is normalized to the total spike count and therefore the two curves can be visually compared more easily. We now also provide statistical analysis based on Poisson statistics that shows that the spike counts between the two groups are significantly different in the time bins immediately following swim onset (Fig. 4g and legend). We also agree that it is a relevant question whether there are positional differences in the tectum with respect to the amount of CD-type inhibition. In this work, however, we limited the recording site to neurons in the central region of the tectal cell body layer. We now display the position of recorded neurons in the tectum (Supplementary Fig. 3d,e) and mention this in the text (page 9, first para). As

these cells cluster within one region, we cannot make a statement on whether there is a gradient of CD inhibition across different tectal regions.

- The authors say that this inhibitory motor related signal is sufficient to shunt visual evoked excitatory inputs from the timing of the two events. However, there seems to be heterogeneity in tectal cells showing the inhibitory current. Did the cells recorded in Fig. 5 also show the inhibitory motor-related current? How do we know that the excitatory input timing is not heterogenous as well? Given this heterogeneity in responses, a direct way to show the shunting would be to record excitatory and inhibitory currents in the same cells and analyze trials with or without motor output.

This is an important aspect. We therefore did more experiments, in which we measured swim-related inhibition and visual motion-evoked excitation in the same cells (Fig. 5d-f). By directly comparing the onset of CD-type inhibition and visually driven excitation within each cell, we found that indeed the inhibition precedes the excitatory input and is therefore appropriately timed to shunt the visually evoked EPSCs, which should effectively suppress activity driven by reafferent-type whole field visual stimulation. We describe the new data in the text (page 9, 2nd para) and show the data in the revised Fig. 5.

- In Fig 6C the temporal resolution of the measurements is low (100ms) relative to the timing differences that are being inferred. Can the authors provide evidence that these inferred timing differences are robust?

We agree that this is a valid question. A higher acquisition speed would of course have been favorable, but we could not increase the frame rate to more than 10 Hz, as it would have come at the cost of spatial resolution, thus limiting our ability to map local Ca²⁺ transients comprehensively across all neuropil layers for individual swims. Nevertheless, our data allowed us to identify local Ca²⁺ transients which showed clear differences in timing relative to the swim onset. To address the question of whether the timing differences are robust, we carefully inspected our data and revised our analysis pipeline substantially in order to make it less model-dependent than the earlier method, where Ca²⁺ signals that were not sufficiently well described by an exponential fit function were excluded. We now reanalyzed the data using a model-free approach to identify Ca²⁺ transients based on evaluating the height and width of peaks relative to the individual noise (standard deviation) of the signal in each ROI (see Methods, page 24, 2nd para; and Supplementary Fig. 4b with legend), which turned out to be more sensitive and identified about 3-times more Ca²⁺ transients near the swim onset than the earlier method (approx 17,000 vs 6,000). To illustrate the heterogeneity of Ca²⁺ transient time courses and differences in timing within the tectal neuropil, we now show the traces from all 172 ROIs measured across the tectal neuropil during a single swim in Supplementary Fig. 4a. Supplementary Fig. 4b explains the method for automatically identifying Ca²⁺ transients. Importantly, we find also in this model-free approach that the timing differences between the different neuropil layers are robust: that is, the strong increase of spontaneous Ca²⁺ signals around the time of swimming (Fig. 6d), with strong peaks in the deep SAC and superficial SM/SO (Fig. 6e), and an overrepresentation of post-swim Ca²⁺ signals in the SM/SO when compared to the distribution across all layers (Fig. 6g). Subsequently, in new experiments in which we imaged TL neurons during spontaneous swimming without visual stimulation, we were able to obtain more direct evidence for the proposed role of TL neurons as a likely relay for feeding CD signals into the tectum via its most superficial layer. We describe these novel results in a new section in the results (page 11, 2nd para) and in a new Fig. 7.

- In Fig 6F which if any of these distributions are statistically different (e.g. using a Kolmogorov-Smirnov test)?

We tested this using Kolmogorov-Smirnov tests, which showed that SAC, SFGS and SM/SO cumulative distributions, but not that for SGC, were different from the cumulative distribution including all layers. This is now stated in the Figure legend (Fig. 6f).

- Fig 7: Why do only some cells receive CD inhibition even though GABA-A labeling is quite diffuse?

We can only speculate on this observation. Possible reasons are that (1) neurons within the same tectal region could be at different stages of maturation; (2) The cells that mediate CD inhibition (putative local inhibitory interneurons that sign-convert the excitatory CD input from TL into a local inhibitory input (proposed model in Fig. 9)) may not be homogeneously distributed; (3) stochastic connectivity patterns might spare some cells from receiving CD inhibition at this developmental stage; (4) GABA-labeling highlights all GABA-ergic synapses, not only those that mediate CD-type inhibition. Other GABAergic synapses could for example come from visually driven superficial interneurons.

- Corollary discharge has been previously shown in the optic tectum of other animals including goldfish, fly, crayfish and mormyrid fish. It would be helpful if the Discussion could include how this study fits into this context of previous findings and what it adds.

To address this point we significantly extended the section on possible functional roles of inhibitory CD signaling in the Discussion (page 16, 1st para, lines 407-419).

- At lines 324-325 the authors say that the CD signal was robust to bout strength. However that's not what they show in Fig 3, and indeed it makes more sense for this dependence to exist.

We agree that this statement was not clear. We rephrased the sentence to emphasize that the motor-related synaptic inhibition occurs during all types of swims and its delay is robust against differences in swim strength and direction (lines 365-67).

Reviewer #2 (Remarks to the Author):

This paper investigates a fundamental issue in sensory neuroscience – how the brain handles sensory inputs that originate from self-motion rather than events in the external world. It has long been suggested that signals mirroring motor commands might project to sensory systems to suppress activity generated during self-motion and such a mechanism has been studied in several contexts, including mechanosensory inputs from the lateral line system of fish that are activated during swimming and visual inputs in *Drosophila* activated during flight. This paper now investigates the visual system of larval zebrafish and demonstrates that the output from the optic tectum is transiently suppressed during “saccade-like” swimming bouts due to activation of local inhibitory neurons. As noted in the Introduction, this observation mirrors the known suppression of visual activity in the superficial layers of the superior colliculus of rodents and primates during saccadic eye movements. This is nonetheless a noteworthy result because it is, I believe, the first direct

demonstration of suppression of visual processing by an efference copy signal in zebrafish. This will in turn open up important possibilities for a dissection of the overall sensory-motor circuits controlling the interaction between motor activity and visual processing, for instance by whole brain imaging at single neuron resolution. The model provided in Fig. 8 provides an overview of the directions in which this research might take.

The work is carried out very carefully and methodically, carefully documenting various properties of the inhibition, such as the delay to its initiation. The consequence of this inhibition on visually-evoked responses is also demonstrated. The data analysis is appropriate and the interpretations are tightly linked to the results. I don't believe that further experiments are required to support the major conclusions. The paper is written clearly and provides the methodological detail for others to repeat the experiments. Together, this work is carried out to a very high standard, providing strong support for the major conclusion that visual input is suppressed during self-motion by inhibition tightly coupled to motor activity.

We thank the reviewer for his/her positive assessment.

Major criticisms

Fig. 7 and associated text is essentially a negative result that does not advance the papers conclusions. I think this investigation of GABA receptor distribution in the tectum could be cut without loss.

Indeed, we agree with this view. It would have been more interesting if there had been layers in the tectum devoid of GABA-A labeling, which would have suggested that CD is unlikely to be transmitted in such layers. As the outcome is negative, we moved this part to the Supplementary Information (Supplementary Fig. 5).

The plot in Fig 3H is hard to see and interpret. I suggest a plot in 2D (x -axis charge, y-axis power) with ipsilateral and contralateral points in different colours.

We agree that Fig. 3h alone makes it difficult to interpret the data. For reasons of space in Fig. 3, we added a new Supplementary Fig. 2, in which the data of panel 3h are now shown for ipsilateral and contralateral data separately in two plots. We hope that this serves the purpose and makes the presentation clearer. In addition, we calculated the Spearman correlation coefficients for the two two-dimensional datasets separately, which yielded the same result as in Fig. 3h: IPSC-charge is significantly correlated with swim-power on the same side of the recorded neuron, but not with that on the opposite side.

Reviewer #3 (Remarks to the Author):

The paper by Ali et al. 'A Synaptic Corollary Discharge Signal Suppresses Midbrain Visual Processing During Saccade-Like Locomotion' describes a midbrain (tectal) mechanism that seems to serve the suppression of visual self-motion signals. Specifically, the authors obtain intracellular (whole-cell) recordings in the tectum of zebrafish larvae. When triggering to the fish self-motion in fictive

swimming they observe an inhibitory signal. Subsequently, the authors show that such inhibition is appropriately timed to suppress visual self-motion-inputs; they also show that this signal is present in visually directed swimming. The authors show that the inhibitory signal enters the visual superficial layers of the zebrafish tectum.

My views on this paper are the following. There is already a lot of evidence from primate and other preparations on the neural mechanisms (Wurtz, Thiele etc) of saccadic suppression, which is appropriately cited by the authors. Despite such prior evidence, I still think that this paper is a big step forward. The reason for this assessment is first, the deep mechanistic analysis by the authors (intracellular recordings and high-resolution imaging), which brings us closer to circuit understanding of self-motion suppression. Second, the quite comprehensive investigation of self-motion-related signals by the authors is decisive, because it convinces the reader that there is really a corollary discharge at work in this preparation. Demonstrating this is an important step for the zebrafish field, because without such evidence we cannot assume that visual self-motion-suppression is at work in this preparation.

We thank the reviewer for his/her positive assessment.

I have only one major point

1) The most interesting question is clearly about the origin of the inhibitory signal. The authors discuss two distinct possible sources of this signal (torus longitudinalis projections or the cerebellum). It is not clear to me, why these two candidate mechanisms could not be differentiated by a quick and dirty experiment such as chopping off the cerebellum. I thought the feasibility of such ad hoc interventions is a major advantage of the zebrafish prep.

This is indeed one of the next big questions. An ablation experiment of the hypothesized pathway components would probably give cues about the sources of the CD signal. Unfortunately, a clean physical ablation experiment to remove the cerebellum or TL is difficult at this stage due to the very small size and compactness of the brain and the lack of well defined borders between adjacent areas. Laser ablations or chemical ablations have been used in the past to assess a causal role of certain cell types or cell clusters in a circuit or behavior, but they are less suitable for anatomically defined, larger regions of the brain, lacking either completeness or specificity. However, to address the question about what pathway components could feed CD signals into the tectum, we performed new experiments, in which we imaged neural activity in the TL during fictive swimming. This provided direct evidence that the TL exhibits swim-related activity temporally locked to spontaneous swim events, in support of its role as a relay for CD signals from premotor areas to the tectum. This is now discussed in a new paragraph in the results section (page 11, 2nd para), a new Fig. 7, and integrated in the model in Fig. 9. To dissect the CD pathway in the zebrafish in further detail, we aim to do more experiments in the future.

Minor

Figure 6 C: showing more nonresponses such as ROI4 would make the figure more interesting and convincing.

This is a valid point. As described above, we have substantially revised our analysis of local Ca²⁺ transients in the tectal neuropil. In the course of this reanalysis we have revised Fig. 6 and show more transients, including non-responsive ROIs. To provide an even more comprehensive overview of non-uniform, swim-related Ca²⁺ signaling, we plotted all ROIs of that example in Supplementary Fig. 4a.

REVIEWERS' COMMENTS

Reviewer #2 (Remarks to the Author):

The authors have carefully revised the manuscript taking into account all the reviewers points. This work is carried out to a very high standard, providing strong support for the conclusion that visual processing in the optic tectum of zebrafish is suppressed by an inhibitory efference copy signal. This work opens up important possibilities for future work analyzing the interaction between motor activity and visual processing, for instance by whole brain imaging at single neuron resolution.

Reviewer #3 (Remarks to the Author):

The authors have largely addressed my concerns. They have not fully identified the source of the corollary signal, but argue persuasively that this question is beyond the scope of the current ms. Also, the responses to the other referees are thorough as far as I can see. I support publication.

Point-by-point response to reviewers' comments

Reviewer #2 (Remarks to the Author):

The authors have carefully revised the manuscript taking into account all the reviewers points. This work is carried out to a very high standard, providing strong support for the conclusion that visual processing in the optic tectum of zebrafish is suppressed by an inhibitory efference copy signal. This work opens up important possibilities for future work analyzing the interaction between motor activity and visual processing, for instance by whole brain imaging at single neuron resolution.

We thank the reviewer for the positive and constructive comments on our work.

Reviewer #3 (Remarks to the Author):

The authors have largely addressed my concerns. They have not fully identified the source of the corollary signal, but argue persuasively that this question is beyond the scope of the current ms. Also, the responses to the other referees are thorough as far as I can see. I support publication.

We thank the reviewer for the positive and constructive comments on our work.